# SIM-CoT: Supervised Implicit Chain-of-Thought

**Xilin Wei**[1,2]**, Xiaoran Liu**[1,4]**, Yuhang Zang**[2✉]**, Xiaoyi Dong**[2]**,**
**Yuhang Cao**[2]**, Jiaqi Wang**[2,4✉]**, Xipeng Qiu**[1,4]**, Dahua Lin**[2,3].
[1] Fudan University, [2] Shanghai AI Laboratory,
[3]The Chinese University of Hong Kong, [4] Shanghai Innovation Institute
xlwei24@m.fudan.edu.cn, zangyuhang@pjlab.org.cn
Github: https://github.com/InternLM/SIM-CoT

## Abstract

Implicit Chain-of-Thought (CoT) methods offer a token-efficient alternative to explicit CoT reasoning in Large Language Models (LLMs), but a persistent performance gap has limited their adoption. We identify a core **latent instability issue** when scaling the computational budget of implicit CoT: as the number of reasoning tokens increases, training often becomes unstable and collapses. Our analysis shows that this instability arises from latent representations becoming homogeneous and losing semantic diversity, caused by insufficient step-level supervision in current implicit CoT methods. To address this, we propose **SIM-CoT**, a plug-and-play training module that introduces step-level supervision to stabilize and enrich the latent reasoning space. SIM-CoT employs an auxiliary decoder during training to align each implicit token with its corresponding explicit reasoning step, ensuring latent states capture distinct and meaningful information. The auxiliary decoder is removed at inference, preserving the efficiency of implicit CoT with no added overhead. It also provides interpretability by projecting each latent token onto an explicit reasoning vocabulary, enabling per-step visualization and diagnosis. SIM-CoT significantly improves both in-domain accuracy and out-of-domain stability of implicit CoT methods, boosting Coconut by +8.2% on GPT-2 and CODI by +3.0% on LLaMA-3.1 8B. It further surpasses the explicit CoT baseline on GPT-2 by 2.1% with 2.3× greater token efficiency, while closing the performance gap on larger models like LLaMA-3.1 8B. Code: https://github.com/InternLM/SIM-CoT.

## 1 Introduction

*"Measure what is measurable, and make measurable what is not so."* — Galileo Galilei

The strong reasoning capabilities of Large Language Models (LLMs) (OpenAI, 2024; Google, 2024; Anthropic, 2024) are often unlocked through explicit Chain-of-Thought (CoT) prompting (Wei et al., 2022). The explicit CoT approach enables LLMs to solve complex problems in a step-by-step manner, yielding high performance in domains like mathematics and programming (Guo et al., 2025; Muennighoff et al., 2025). Despite its advantages, explicit CoT also faces several limitations. For instance, explicit CoT approaches must verbalize intermediate thoughts from a fixed vocabulary, thereby precluding the exploration of alternative solution paths (Li et al., 2025; Zhang et al., 2025). Additionally, the generation of extensive intermediate sequences significantly increases inference cost and can result in redundant over-thinking steps or unnecessary verbosity (Chen et al., 2024).

To address the flexibility and efficiency issues of explicit CoT methods, recent **implicit CoT** approaches (Hao et al., 2025; Zhang et al., 2025; Li et al., 2025) have been proposed by representing reasoning in a continuous latent space rather than as a sequence of discrete text tokens. The implicit CoT methods allow each latent representation to encode richer information than a single explicit token, often with a significantly smaller number of latents than the length of an explicit reasoning chain. Early representative implicit work like Coconut (Hao et al., 2025) improves efficiency while still capturing useful intermediate structure. More recent approaches, such as CODI (Shen et al., 2025), further apply trajectory-level distillation from explicit reasoning paths to enhance performance.

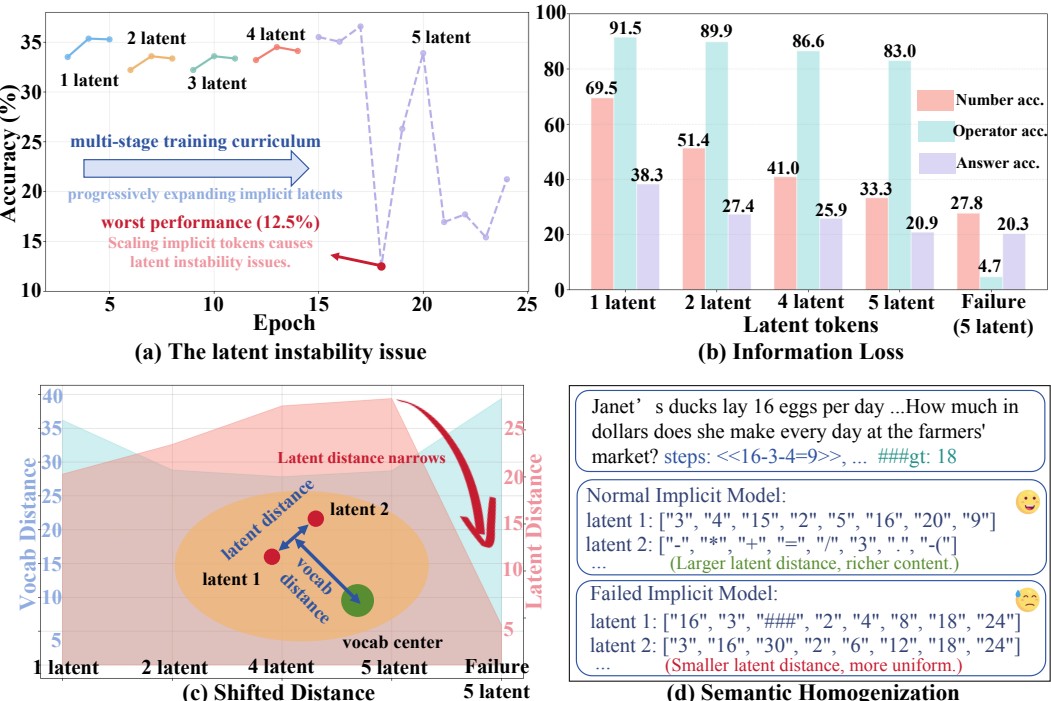

Figure 1: **(a) The latent instability issue**: while using more implicit tokens initially improves accuracy, training becomes unstable and sometimes collapses. **(b) Information Loss**: the implicit tokens of failed models (5 latent tokens) lose crucial information about operators (like $+$, $-$), which makes complex reasoning impossible. **(c) Shifted Distance**: the latent-to-latent distance of failed models shrinks and becomes too similar to each other, while the latent drifts away from the central vocabulary embedding space. **(d) Semantic Homogenization**: failed models produce similar latent representations, resulting in a narrower range of decoded tokens, mostly numbers, as opposed to the more varied content generated by a normal model.

Despite these advancements, a **performance gap** still exists between existing implicit CoT methods and their explicit counterparts. The implicit CoT approaches are *fast*, *token-efficient* but *less accurate*, which currently limits their broader application.

To narrow the performance gap, inspired by the success of explicit CoT that scales computational budget for better performance, we explore a similar strategy for implicit CoT methods by increasing the number of implicit tokens. However, in Fig. 1 **(a)**, we reveal one underlying **latent instability issue** in current implicit CoT approaches. As we extend the number of implicit tokens from the default three (Hao et al., 2025) to five, the training process initially improves accuracy but becomes unstable and sometimes collapses entirely. To interpret the **latent instability issue**, we analyze implicit tokens from models trained on math reasoning data GSM8K-Aug (Deng et al., 2024). We follow previous works (Hao et al., 2025; Deng et al., 2024) to project the implicit tokens through the LM head and examine their top decoded tokens for analysis. As shown in Fig. 1 **(b)**, failed models tend to collapse into homogeneous latent states. While successful reasoning requires capturing both numerical and operator information, the implicit tokens of failed models primarily represent numbers, almost completely losing the critical operator information. Fig. 1 **(c)** further demonstrates that a model's collapse is accompanied by two changes: a reduction in the inter-latent distance and a drift of the latent states away from the central vocabulary embedding space. The latent representations of failed models become too similar and lose their semantic connection to the tokens they are meant to represent. Fig. 1 **(d)** provides an example of the semantic homogenization. A normal model (top) maintains a large distance between its two latent tokens, allowing them to capture distinct information for numbers and operators. In contrast, a failed model (bottom)'s latent tokens become homogeneous, with both states decoding to similar information, primarily numbers.

Our observation (Fig. 1) reveals the reasons for the latent instability issue: a lack of sufficient step-level supervision for existing implicit methods to maintain the rich and varied internal representations.

Without stronger guidance, the latent space collapses, losing its diversity and making it impossible to reliably encode the distinct, step-level reasoning needed for complex reasoning tasks. Motivated by our findings, we propose **S**upervised **IM**plicit-CoT (**SIM-CoT**), a plug-and-play module that introduces step-level supervision for implicit CoT approaches to alleviate the latent instability issue. Instead of supervising only the final answer (Hao et al., 2025) or the trajectory (Shen et al., 2025), SIM-CoT uses an auxiliary decoder to align each implicit token with its corresponding explicit reasoning step during training. The step-level supervision for implicit tokens stabilizes optimization, prevents collapse, and ensures that latent tokens capture meaningful reasoning content. Crucially, because the auxiliary decoder is removed during inference, our approach incurs virtually no extra computational cost, making it as efficient as standard implicit CoT approaches. Beyond *accuracy*, *stability*, and *efficiency*, the auxiliary decoder also affords *interpretability* of implicit reasoning. During training, it defines a projection from latent tokens to the explicit reasoning vocabulary, enabling us to decode each latent step into a human-interpretable summary for verification or error diagnosis.

Experiments show that SIM-CoT acts as a plug-and-play module that boosts both accuracy and stability. We show that SIM-CoT can be effortlessly combined with various implicit CoT approaches such as Coconut (Hao et al., 2025), CODI (Shen et al., 2025), and training-free approaches (Zhang et al., 2025) to further enhance reasoning performance. On GPT-2, SIM-CoT surpasses both the strong explicit baseline (supervised fine-tuning on explicit CoT data) by 2.1%, and outperforms existing implicit methods Coconut and CODI by 8.2% and 4.3%, respectively. The performance trend holds as the method scales to larger models such as the LLaMA series. SIM-CoT achieves improvements over CODI of 3.4% (LLaMA-3.2 1B), 1.5% (LLaMA-3.2 3B), and 3.0% (LLaMA-3.1 8B), in addition to a 9.0% gain over Coconut on the LLaMA-3.2 1B model. Furthermore, while previous implicit CoT approaches (e.g., Coconut) collapse when scaled to 8 or 16 implicit tokens, SIM-CoT remains stable and continues to boost performance.

In summary, our contributions are as follows: **1)** We provide a systematic analysis of the latent instability issue of implicit CoT approaches, showing that instability and collapse arise from insufficient supervision. **2)** We introduce SIM-CoT, which applies step-level supervision to the model's implicit tokens. SIM-CoT not only integrates seamlessly with existing implicit CoT approaches and boosts performance with minimal inference overhead, but also affords interpretability of implicit reasoning by projecting each latent token onto an explicit reasoning vocabulary, enabling per-step visualization of semantic roles and diagnosis. **3)** Through extensive experiments, we demonstrate that SIM-CoT not only improves accuracy in the in-domain dataset, but also generalizes effectively to out-of-domain datasets. The performance gains are consistent across a range of LLMs, including GPT-2 and recent LLaMA 3 models (1B, 3B, and 8B).

## 2 ANALYSIS OF IMPLICIT COT: THE LATENT INSTABILITY ISSUE

We first present an analysis (Fig. 1) of the limitations in implicit latent CoT approaches. We follow Coconut (Hao et al., 2025) and analyze implicit latents by projecting them through the LM head and examining the top-8 decoded tokens to understand the semantic and geometric properties.

**Latent Instability Issue.** Fig. 1 **(a)** shows the training process of Coconut when the number of implicit latent tokens is progressively increased. Initially, as the number of latents increases from one to four, the model's accuracy generally improves, suggesting that using more latents can enhance performance. However, a significant drop in accuracy occurs when the number of latents is scaled to five, with performance collapsing to its worst point of 12.5%. The latent instability issue indicates that the implicit reasoning approach is sensitive to the choice of the number of latent tokens, as shown by the sharp drop and subsequent fluctuations in accuracy after adding the fifth latent.

**Information Loss.** Fig. 1 **(b)** presents an analysis of how different levels of accuracy are affected by the number of latent tokens, using accuracy metrics at three levels: number, operator, and answer. The bar chart reveals a clear trend: as the number of latent tokens increases from 1 to 5, there is a general decline in performance across all three metrics, especially for the operator accuracy. The strong correlation between increased latent tokens and declining performance, particularly the sharp fall during failure, suggests that implicit latents do not consistently capture the necessary compositional reasoning process without more explicit, fine-grained supervision.

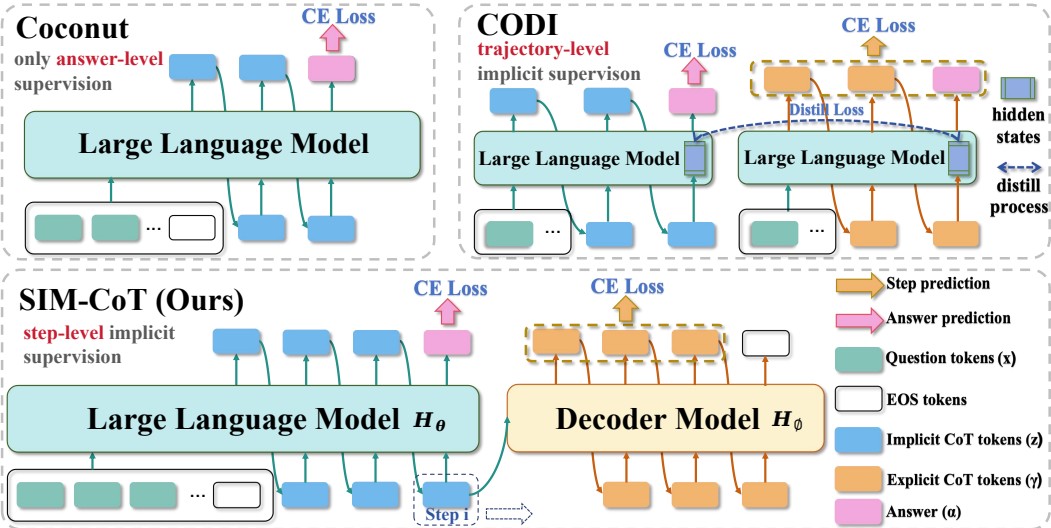

Figure 2: The framework comparison between Coconut (upper left), CODI (upper right), and our SIM-CoT (bottom). Unlike Coconut and CODI, which apply coarse-grained supervision on answers or trajectories, our **SIM-CoT** employs a decoder to **align implicit latents with step-level reasoning**, enhancing performance while maintaining inference efficiency.

**Shifted Distance.** Fig. 1 **(c)** examines the geometric properties of the latent representations during training. Two metrics are analyzed: the Latent Distance (red), which measures the average distance between pairs of latent vectors, and the Vocab Distance (blue), which measures the average distance from each latent vector to the center of the vocabulary embedding space. When the latent CoT model collapses, the latent distance decreases sharply, indicating that the latent vectors are collapsing and becoming nearly identical, losing their distinctiveness. Simultaneously, the vocab distance increases, showing that these collapsing latents are drifting away from the main lexical embedding space and are no longer grounded in the fundamental token representations used by the model.

**Semantic Homogenization.** Fig. 1 **(d)** provides a qualitative analysis of the content of the latent tokens in a normal case versus a failed model. In the normal implicit model (middle), the decoded tokens from the latents are diverse and meaningful. In the failed implicit model (bottom), the semantic content of the latents becomes highly homogeneous. Latent 1 and Latent 2 contain mainly numbers, lacking operators or symbolic information needed for calculation. This shows that successful training produces latents with step-wise reasoning, while without explicit supervision, the latent space collapses into uniform numerical forms.

**Summary.** Our analysis across Fig. 1 (a-d) highlights a crucial trade-off between diversity and stability. When the model collapses, it loses both its diversity (as the latents become too similar) and its stability (as the latents move away from the token space), leading to catastrophic information loss and a complete failure of the reasoning process, as shown by the sharp drop in overall accuracy. These combined findings show that without proper guidance, the latent space degenerates, losing its ability to represent distinct reasoning steps. These challenges motivate our proposed method, which introduces **step-level implicit supervision** to stabilize the training process and enrich unique semantic content of each latent, all while maintaining efficiency during inference.

## 3 METHODOLOGY

**Overview.** As shown in Fig. 2, early implicit reasoning studies differ mainly in supervision granularity: **Coconut** (top left) uses answer-level supervision, while **CODI** (top right) introduces trajectory-level signals via distillation. Both remain coarse and do not tell the model which latent should encode which step. We propose **SIM-CoT**, which provides **step-level implicit supervision**: During an **implicit phase**, the LLM runs for a fixed number $K$ of reasoning steps; at each step $k$ it takes the **last hidden state** as the implicit latent $z_k$ and appends it to the sequence as the next "token" vector. After $K$ steps, the model switches back to **explicit** decoding over the vocabulary to generate the

final answer. A decoder is used only in training to align each $z_k$ with the textual content of the $k$-th reasoning step; at inference, the decoder is removed, so the runtime is essentially that of direct answer generation plus $K$ forward positions, which is far shorter than explicit CoT token lengths.

## 3.1 NOTATION

Let $\mathcal{V}$ be the vocabulary and $E \in \mathbb{R}^{|\mathcal{V}| \times d}$ the token embedding matrix. A question is $x = (x_1, \ldots, x_T) \in \mathcal{V}^T$ with embedded prefix

$$U^{(0)} = \big(e(x_1), \ldots, e(x_T)\big), \quad e(\cdot) \in \mathbb{R}^d.$$

We run an autoregressive LLM $F_\theta$ on any prefix $U = (u_1, \ldots, u_m)$ of $d$-dimensional vectors (tokens or latents). Denote the last-layer hidden state at the final position by

$$H_\theta(U) \in \mathbb{R}^d.$$

For supervision, the $k$-th textual step is $s_k = (y_{k,1}, \ldots, y_{k,L_k}) \in \mathcal{V}^{L_k}$, and the answer is $a = (a_1, \ldots, a_{L_a}) \in \mathcal{V}^{L_a}$. The auxiliary decoder has parameters $\phi$; the LLM has parameters $\theta$.

## 3.2 IMPLICIT PHASE: LATENT CONSTRUCTION BY LAST HIDDEN STATES

We fix the number of implicit reasoning steps $K$ in advance. For each step $k = 1, \ldots, K$,

$$z_k = H_\theta\big(U^{(k-1)}\big) \in \mathbb{R}^d, \qquad U^{(k)} = U^{(k-1)} \oplus z_k, \tag{1}$$

where $\oplus$ denotes concatenation along the time axis. The implicit chain-of-thought is therefore represented as a continuous sequence of hidden states $z_{1:K} = (z_1, \ldots, z_K)$, which are autoregressively generated and appended to the context before the model switches to explicit decoding.

## 3.3 EXPLICIT PHASE: ANSWER DECODING OVER THE VOCABULARY

After constructing the implicit latents $z_{1:K}$, the model switches to explicit decoding to generate the final answer. Let $W_o \in \mathbb{R}^{|\mathcal{V}| \times d}$ be the output projection (LM head). With teacher forcing on the partial answer $a_{<t}$, the generation is

$$h_{T+K+t} = H_\theta\big(U^{(K)} \oplus e(a_{<t})\big), \tag{2}$$

$$p_\theta(a_t \mid x, z_{1:K}, a_{<t}) = \text{softmax}\big(W_o \, h_{T+K+t}\big)_{a_t}, \tag{3}$$

$$p_\theta(a \mid x, z_{1:K}) = \prod_{t=1}^{L_a} p_\theta(a_t \mid x, z_{1:K}, a_{<t}). \tag{4}$$

## 3.4 TRAINING-TIME DECODER AND STEP-LEVEL SUPERVISION

During training, a decoder $p_\phi$ (architecturally identical to the LLM) takes only the $k$-th implicit latent $z_k$ as conditioning signal and autoregressively generates the $k$-th textual step $s_k = (y_{k,1}, \ldots, y_{k,L_k})$. This provides **step-level** supervision that directly grounds $z_k$ to its corresponding reasoning content:

$$p_\phi(s_{1:K} \mid z_{1:K}) = \prod_{k=1}^{K} p_\phi(s_k \mid z_k) = \prod_{k=1}^{K} \prod_{t=1}^{L_k} p_\phi\big(y_{k,t} \mid z_k, \, y_{k,<t}\big). \tag{5}$$

*Parameterization.* For step $k$, the decoder is conditioned on the implicit latent $z_k$ obtained from the LLM. Since $z_k$ does not correspond to any token in the vocabulary, it is not included in the loss calculation. Instead, $z_k$ is injected as an additional prefix vector that initializes the decoder's hidden state for step generation. Concretely, the decoder input sequence is

$$U_k^{\text{dec}} = \big[\, z_k \,;\, e(y_{k,1}), \ldots, e(y_{k,L_k}) \,\big],$$

where $e(\cdot)$ denotes the embedding function of the LLM shared between both models. During training with teacher forcing, the decoder predicts each token $y_{k,t}$ autoregressively:

$$p_\phi(y_{k,t} \mid z_k, y_{k,<t}) = \text{softmax}\big(W^{\text{dec}} h_{k,t}^{\text{dec}}\big)_{y_{k,t}},$$

where $h_{k,t}^{\text{dec}}$ is the decoder hidden state at position $t$ and $W^{\text{dec}}$ is the LM head of the decoder.

The training loss for step $k$ is then

$$\mathcal{L}_{\text{step},k} = -\sum_{t=1}^{L_k} \log p_\phi(y_{k,t} \mid z_k, y_{k,<t}),$$

which supervises only the textual step tokens. The decoder is used exclusively for this supervision during training and is discarded at inference.

## 3.5 Objectives

Training involves two complementary cross-entropy losses: one for supervising the textual steps through the decoder, and one for supervising the final answer through the base LLM.

**Step-level supervision.** For each implicit latent $z_k$, the decoder $p_\phi$ generates the corresponding reasoning step $s_k = (y_{k,1}, \ldots, y_{k,L_k})$. Since $z_k$ is not a vocabulary token, the loss is computed only over the textual step tokens:

$$\mathcal{L}_{\text{step}} = -\sum_{k=1}^{K} \sum_{t=1}^{L_k} \log p_\phi\big(y_{k,t} \mid z_k, y_{k,<t}\big). \tag{6}$$

This loss grounds each latent $z_k$ to a specific reasoning step, ensuring that the latent sequence carries fine-grained semantics.

**Answer supervision.** After $K$ implicit steps, the LLM $F_\theta$ switches back to explicit decoding to generate the final answer $a = (a_1, \ldots, a_{L_a})$. We optimize the standard language modeling loss:

$$\mathcal{L}_{\text{ans-lm}} = -\sum_{t=1}^{L_a} \log p_\theta\big(a_t \mid x, z_{1:K}, a_{<t}\big). \tag{7}$$

**Total objective.** The overall loss is a weighted sum:

$$\mathcal{L} = \lambda_{\text{step}} \, \mathcal{L}_{\text{step}} + \lambda_{\text{lm}} \, \mathcal{L}_{\text{ans-lm}}. \tag{8}$$

Gradients from $\mathcal{L}_{\text{step}}$ propagate through the decoder into the latent representations $z_{1:K}$ and further into the LLM (via Eq. equation 1), shaping the hidden states to encode step-level reasoning. Meanwhile, $\mathcal{L}_{\text{ans-lm}}$ trains the base model to produce the final answer directly, so the decoder can be discarded at inference time without affecting efficiency. Implementation details, inference procedures, and diagnostic analyses are provided in Appendix D.

## 4 Experiment

### 4.1 Experimental Setup

**Training Data.** We follow previous works (Deng et al., 2024; Hao et al., 2025) to use the **GSM8k-Aug** dataset Deng et al. (2024) for training implicit CoT models. The GSM8k-Aug expands the original GSM8k training set (Cobbe et al., 2021) to 385k examples by using GPT-4 for data generation. To facilitate implicit CoT training, the GSM8k-Aug removes the reasoning chain of natural language, preserving only a sequence of structured mathematical expressions. Each expression is logically linked to the previous step, as illustrated by the example: `<<12*3=36>><<9*2=18>><<17*2=34>><<36+18+34=88>>`.

**Evaluation Benchmarks.** We report results on the **GSM8k-Aug** test set (Cobbe et al., 2021), which serves as our in-domain (ID) evaluation benchmark. To further evaluate mathematical reasoning under a distribution shift, we also evaluate models on three out-of-domain (OOD) benchmarks: (1) **SVAMP** (Patel et al., 2021), a dataset of grade-school arithmetic word problems that introduces simple variations to assess robustness; (2) **GSM-Hard** (Gao et al., 2022), a modified version of the GSM8k test split where numbers are replaced with larger magnitudes to increase problem difficulty; and (3) **MultiArith** (Roy & Roth, 2015), a subset of MAWPS (Koncel-Kedziorski et al., 2016) consisting of multi-step arithmetic word problems. Please refer to the Appendix D for more details.

Table 1: **Main results on GPT-2**. We report accuracy (%) on *in-domain* (GSM8k-Aug) and *out-of-domain* (GSM-Hard, MultiArith, SVAMP) benchmarks. Our SIM-CoT is shown to provide accuracy gains on top of existing methods such as Coconut (Hao et al., 2025) and CODI (Shen et al., 2025).

| Method | SIM-CoT | In-domain | | Out-of-domain | | | | |
| | | GSM8k-Aug | | GSM-Hard | MultiArith | SVAMP | Average | # Average |
| | | Acc. (%) | # Tokens | Acc. (%) | Acc. (%) | Acc. (%) | Acc. (%) | Tokens |
|---|---|---|---|---|---|---|---|---|
| SFT-CoT | ✗ | 42.7 | 27.6 | 9.0 | 85.0 | 41.6 | 45.2 | 24.7 |
| No-CoT | ✗ | 19.1 | 2.2 | 4.3 | 41.1 | 16.4 | 20.6 | 1.4 |
| iCoT | ✗ | 30.1 | 2.2 | 5.7 | 55.5 | 29.4 | 30.2 | 1.4 |
| Coconut | ✗ | 36.6 | 12.2 | 8.1 | 83.5 | 36.2 | 42.6 | 11.4 |
| | ✓ | **44.8** (+8.2) | 12.2 | **9.3** | **90.8** | **40.7** | **46.9** (+4.3) | 11.4 |
| CODI | ✗ | 42.0 | 12.2 | **9.4** | **93.0** | 41.7 | 48.0 | 12.6 |
| | ✓ | **42.6** (+0.6) | 12.2 | **9.4** | 92.8 | **42.6** | **48.3** (+0.3) | 12.6 |

Table 2: **Main results on LLaMA 3.2 1B.** We report accuracy (%) on *in-domain* (GSM8k-Aug) and *out-of-domain* (GSM-Hard, MultiArith, SVAMP) benchmarks. Our SIM-CoT builds on CODI to achieve a new SOTA in implicit reasoning while setting performance comparable to explicit CoT.

| Method | SIM-CoT | In-domain | | Out-of-domain | | | | |
| | | GSM8k-Aug | | GSM-Hard | MultiArith | SVAMP | Average | # Average |
| | | Acc. (%) | # Tokens | Acc. (%) | Acc. (%) | Acc. (%) | Acc. (%) | Tokens |
|---|---|---|---|---|---|---|---|---|
| SFT-CoT | ✗ | 58.4 | 25.3 | 13.9 | 96.7 | 65.7 | 58.8 | 23.1 |
| No-CoT | ✗ | 28.8 | 1.2 | 6.3 | 50.3 | 26.7 | 27.8 | 1.9 |
| iCoT | ✗ | 19.0 | 1.2 | 4.4 | 39.0 | 40.9 | 28.1 | 1.9 |
| Coconut | ✗ | 33.2 | 13.2 | 7.0 | 63.3 | 43.7 | 38.0 | 11.9 |
| | ✓ | **42.2** (+9.0) | 13.2 | **9.3** | **87.7** | **43.9** | **47.0** (+9.0) | 11.9 |
| CODI | ✗ | 52.7 | 13.2 | 11.9 | 95.0 | 60.6 | 55.8 | 13.4 |
| | ✓ | **56.1** (+3.4) | 13.2 | **12.7** | **96.2** | **61.5** | **56.8** (+1.0) | 13.4 |

**Implementation Details.** We follow the training setup of previous works (Hao et al., 2025; Shen et al., 2025), and adopt consistent hyperparameter choices for GPT-2, LLaMA 1B/3B/8B. Detailed configurations, such as learning rates, curriculum strategies, are provided in Appendix C.

## 4.2 MAIN RESULTS

**Baselines.** We compare our SIM-CoT against five representative baselines: (1) **CoT-SFT**: Supervised fine-tuning (SFT) on CoT-annotated data, where the model is trained to generate explicit intermediate reasoning steps followed by the final answer. (2) **No-CoT-SFT**: Supervised fine-tuning on direct answers only, without producing intermediate steps. (3) **iCoT** (Deng et al., 2024): A curriculum learning method based on "Stepwise Internalization," which injects CoT reasoning patterns into the model's internal representations, enabling it to produce more accurate direct answers during inference. (4) **Coconut** (Hao et al., 2025): A curriculum learning approach that gradually replaces explicit reasoning steps with implicit tokens until the reasoning process becomes fully implicit. This method has shown strong empirical performance and serves as a primary baseline in our experiments. (5) **CODI** (Shen et al., 2025): A distillation-based method where explicit CoT acts as the teacher and implicit CoT as the student. By aligning the last hidden states of the full reasoning trajectory, CODI effectively internalizes knowledge and alleviates catastrophic forgetting.

**In-Domain Math Benchmark Results.** Table 1 (first column) reports GPT-2 results on GSM8k-Aug. SIM-CoT outperforms SFT-CoT and is the first training-based approach where implicit CoT surpasses explicit CoT. With GPT-2 using Coconut as the backbone, it achieves a $+2.1$ point improvement over SFT-CoT. It also exceeds other training-based implicit reasoning models; for example, on Coconut, it improves by $+8.2$ points, a relative gain of $22.4\%$. Moreover, when applied on top of CODI—the current SOTA implicit reasoning method—SIM-CoT yields an additional $+0.6$ point improvement.

Table 2 (first column) shows the results when CODI is used as the backbone. In this setting, our method achieves a substantial $+3.4$ point improvement. Furthermore, we are the first to achieve performance comparable to SFT-CoT on LLaMA-1B, reaching $96\%$ of its accuracy. Given that prior studies (Xu et al., 2025; Shen et al., 2025) reported that curriculum learning in larger models leads to catastrophic forgetting and that homogeneous knowledge harms model training (Zhao et al., 2023),

Table 3: Main results on larger LLaMA models (3B and 8B). We report accuracy (%) on **in-domain** (GSM8k-Aug) and **out-of-domain** (GSM-Hard, MultiArith, SVAMP) benchmarks.

| Model | Method | SIM-CoT | In-domain | | Out-of-domain | | | | |
|-------|--------|---------|-----------|----------|-----------|------------|---------|---------|-----------|
| | | | GSM8k-Aug | | GSM-Hard | MultiArith | SVAMP | Average | # Average |
| | | | Acc. (%) | # Tokens | Acc. (%) | Acc. (%) | Acc. (%) | Acc. (%) | Tokens |
| LLaMA 3.2 3B | SFT-CoT | ✗ | 71.5 | 27.7 | 17.0 | 98.3 | 71.0 | 62.1 | 22.4 |
| | No-CoT | ✗ | 38.3 | 1.2 | 9.5 | 88.7 | 52.9 | 50.4 | 1.4 |
| | CODI | ✗ | 60.8 | 7.2 | 14.3 | 98.7 | 73.3 | 62.1 | 7.5 |
| | | ✓ | **62.3** (+1.5) | 7.2 | **14.6** | **98.8** | **74.9** | **62.8** (+0.7) | 7.5 |
| LLaMA 3.1 8B | SFT-CoT | ✗ | 71.7 | 27.4 | 16.5 | 98.3 | 73.1 | 62.6 | 22.2 |
| | No-CoT | ✗ | 39.5 | 1.2 | 9.8 | 88.0 | 55.3 | 51.0 | 1.6 |
| | CODI | ✗ | 61.1 | 7.2 | 15.5 | 99.5 | 78.1 | 64.4 | 7.5 |
| | | ✓ | **64.1** (+3.0) | 7.2 | **16.3** | **100.0** | **79.4** | **65.2** (+0.8) | 7.5 |

Figure 3: Ablation study on different numbers of implicit latents. The x-axis denotes the number of implicit latents and implicit tokens (joined with "-"), while the y-axis denotes accuracy. The blue line corresponds to our method SIM-CoT, and the orange line corresponds to the baseline Coconut.

we choose CODI as the backbone because its KL-regularized objective constrains the training not to deviate too far from the original model distribution, thereby alleviating catastrophic forgetting.

**Out-of-Domain Math Benchmark Results.** To evaluate the robustness of our method, we train on GSM8k and evaluate on out-of-domain datasets (GSM-Hard, MultiArith, and SVAMP). From the third column of Table 1, we observe that SIM-CoT consistently outperforms SFT-CoT, with an average improvement of +4.3 points when using Coconut as the backbone. From the third column of Table 2, our method further improves upon the current SOTA implicit reasoning method CODI by +1.0 point. Moreover, when scaling model size from GPT-2 to LLaMA-1B, SIM-CoT enlarges the performance gap against iCoT, Coconut, and other baselines.

We attribute the robustness of SIM-CoT to its step-level implicit supervision. Unlike SFT-CoT, which forces the model to mimic deterministic natural language annotations, and unlike CODI, which applies trajectory-level alignment to a coarse-grained reasoning path, our method introduces a moderate form of supervision. This design ensures the plausibility of each reasoning step while preserving the diversity of reasoning trajectories, thereby improving generalization to unseen inputs.

**Inference Efficiency.** In terms of inference speed, our method maintains the same efficiency as other implicit reasoning approaches on both GPT-2 and LLaMA-1B. On GPT-2, SIM-CoT not only surpasses SFT-CoT on both in-domain and out-of-domain benchmarks, but also achieves a $2.3\times$ and $2.2\times$ speedup on Coconut, respectively. On LLaMA-1B, SIM-CoT remains comparable to SFT-CoT in accuracy while delivering $1.9\times$ and $1.7\times$ speedups on in-domain and out-of-domain benchmarks, respectively. These results demonstrate the effectiveness of our approach in retaining or even enhancing the performance of explicit CoT while substantially reducing inference cost.

## 4.3 ABLATION STUDIES

**Ablation on the Number of Implicit Tokens.** We study the effect of varying the number of implicit latents on GPT-2, comparing SIM-CoT with Coconut trained on GSM8k-Aug and evaluated on GSM8k-Aug, GSM-Hard, MultiArith, and SVAMP (Fig. 3). Following Coconut, each latent corresponds to two tokens. As shown in Fig. 5, most problems involve two to six steps with a small proportion of harder cases, so we set the maximum number of implicit latents to 8. For each configuration, we report the best performance, and results show that SIM-CoT provides more stable training and achieves consistent gains over Coconut, indicating that step-level implicit supervision scales effectively with larger latent capacity.

Table 4: Comparison of (a) LLaMA 1B with different decoders and (b) latent token distance analysis. In (a), we evaluate the effect of using larger decoders with a 1B model on both in-domain (GSM8k-Aug) and out-of-domain benchmarks (GSM-Hard, MultiArith, SVAMP). In (b), we report average pairwise distances among latent tokens (Dist.) and their distances to the vocabulary center (Dist. to VC) under different settings, including failed cases and the effect after applying SIM-CoT.

(a) LLaMA 1B with different decoders.

| Model | In-domain | Out-of-domain | | |
|---|---|---|---|---|
| | GSM8k-Aug | GSM-Hard | MultiArith | SVAMP |
| Baseline | 52.7 | 11.9 | 95.0 | 60.6 |
| + 1B Decoder | **56.1** | **12.7** | **96.2** | **61.5** |
| + 3B Decoder | 50.4 | 11.6 | 95.6 | 59.8 |
| + 8B Decoder | 50.0 | 11.7 | 94.2 | 56.8 |

(b) Latent token distance analysis.

| Setting | Dist. | Dist. to VC |
|---|---|---|
| 1 latent | 20.30 | 36.20 |
| 2 latent | 23.46 | 28.82 |
| 4 latent | 27.56 | 27.83 |
| 5 latent | 28.34 | 28.34 |
| Fail 5 latent | 4.21 | 39.39 |
| After SIM-CoT | 32.81 | 29.80 |

Table 5: Ablation study of soft thinking on LLaMA 3.2 1B. We report accuracy (%) on the in-domain dataset (GSM8k-Aug) and out-of-domain datasets (GSM-Hard, MultiArith, and SVAMP). Adding soft thinking consistently improves both Coconut and SIM-CoT across all benchmarks, showing its effectiveness in enhancing implicit reasoning.

| Method | GSM8k-Aug | GSM-Hard | MultiArith | SVAMP |
|---|---|---|---|---|
| Coconut | 36.6 | 8.1 | 83.5 | 36.2 |
| + Soft Thinking | 36.7 | 8.3 | 85.2 | 36.0 |
| SIM-CoT | 44.8 | 9.3 | 90.8 | 40.7 |
| + Soft Thinking | **45.0** | **9.4** | **91.5** | **40.8** |

**Ablation on Scaling to Larger Backbones.** To examine robustness and scalability, we extend experiments to larger LLaMA backbones, including LLaMA 3.2 3B and LLaMA 3.1 8B. Table 3 reports results on GSM8k-Aug (in-domain) and GSM-Hard, MultiArith, and SVAMP (out-of-domain).

Overall, SIM-CoT scales effectively to larger backbones, consistently surpassing or matching explicit CoT on out-of-domain tasks while reducing reliance on trajectory-level supervision.

On **LLaMA 3.2 3B**, SIM-CoT improves over CODI by $+1.5$ points on GSM8k-Aug and $+1.6$ points on SVAMP, while maintaining comparable performance on GSM-Hard and MultiArith. This demonstrates that step-level implicit supervision strengthens strong implicit reasoning baselines even at larger scales.

On **LLaMA 3.1 8B**, SIM-CoT yields gains of $+3.0$ points on GSM8k-Aug, $+1.3$ on SVAMP, and $+0.8$ on MultiArith relative to CODI, while maintaining stable accuracy on GSM-Hard. Compared with SFT-CoT, it achieves higher accuracy on MultiArith (100.0 vs. 98.3) and SVAMP (79.4 vs. 73.1), while remaining similar on GSM-Hard.

Together, these results confirm that SIM-CoT scales effectively to larger backbones, providing consistent gains across both in-domain and out-of-domain benchmarks with reduced reliance on trajectory-level supervision.

**Ablation on Different Decoder Sizes.** We investigate how decoder size affects performance by replacing the decoder of the LLaMA 1B backbone with larger versions from the same vocabulary family and evaluating on GSM8k-Aug, GSM-Hard, MultiArith, and SVAMP. As shown in Table 4(a), integrating a 1B-scale decoder leads to consistent improvements across all benchmarks. However, simply scaling the decoder to larger variants (3B or 8B) does not yield additional benefits and instead slightly reduces accuracy.

These results suggest that moderate decoder scaling can enhance reasoning ability, but excessively large decoders may introduce optimization challenges or misalignment with the 1B backbone, ultimately limiting generalization. A plausible explanation is that the 1B encoder and 1B decoder originate from the same model family and thus share a more compatible representation space, facilitating stable learning. In contrast, larger decoders (3B or 8B) may require implicit projection to align with the 1B backbone, which can introduce representational mismatches and hinder training stability.

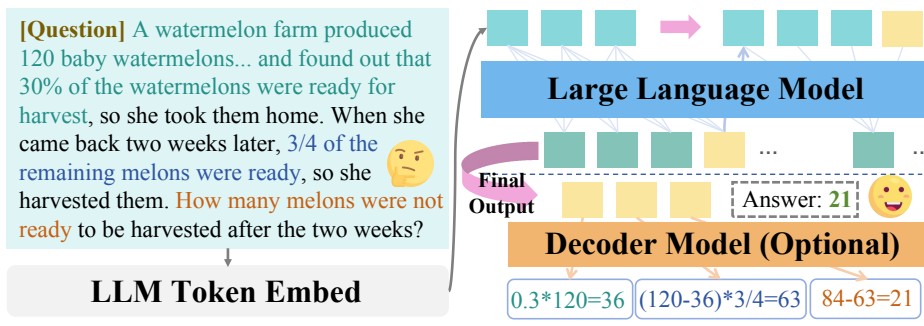

Figure 4: SIM-CoT case study on GSM8k. The generated implicit continuous tokens are subsequently interpreted by our decoder, which visualizes the solution intermediate steps leading to the final output.

**Ablation on Soft Thinking.** We also study the effect of integrating soft thinking (Zhang et al., 2025; Wu et al., 2025) with both Coconut and SIM-CoT. For clarity, the detailed experimental setup, results, and analyses are provided in Appendix A.

**Interpretability of Implicit Reasoning.** Implicit reasoning models generate continuous latent thoughts that are not directly human-readable. We reuse the training decoder to project each latent step into text space, enabling per-step semantic visualization (Fig. 4; Appendix G).

To analyze the latent space, we examine two geometric measures: average pairwise distance (Dist.) and distance to the vocabulary center (Dist. to VC) (Table 4(b), Fig. 6). As latent tokens increase from 1 to 5, Dist. grows (20.30→28.34), indicating better separability; in the failed 5-latent case, it collapses (4.21), while SIM-CoT restores and further enlarges it (32.81). Dist. to VC decreases from 36.20 to 28 with more tokens, suggesting improved alignment, but spikes to 39.39 in the failed case; SIM-CoT stabilizes it at 29.80. Qualitative results confirm these trends: normal tokens remain separated and grounded, failed tokens collapse and drift, and SIM-CoT recovers a structured, stable latent space.

## 5 RELATED WORK

A large body of work has studied explicit chain-of-thought (CoT) prompting, including self-consistency (Wei et al., 2022; Wang et al., 2023), least-to-most prompting (Zhou et al., 2023), reflection-based reasoning (Shinn et al., 2023; Madaan et al., 2023), and the integration of external tools (Yao et al., 2023). Other work investigates step-level supervision to structure explicit reasoning (Zheng et al., 2023; Wei et al., 2025). While effective, explicit CoT increases inference cost with longer sequences and often produces redundant steps, limiting efficiency and reasoning diversity (Li et al., 2025; Zhang et al., 2025; Xu et al., 2025).

Implicit CoT aims to reduce output length while retaining multi-step reasoning. Prior work explores knowledge internalization (Deng et al., 2024), architectural modification (Saunshi et al., 2025; Chen et al., 2025; Cheng & Van Durme, 2024; Su et al., 2025; Mohtashami et al., 2023; Geiping et al., 2025), training-free latent construction (Zhang et al., 2025; Wu et al., 2025), and auto-regressive latent reasoning (Xu et al., 2025; Tan et al., 2025). Coconut applies answer-level supervision (Hao et al., 2025), and CODI uses trajectory-level distillation (Shen et al., 2025). Our work introduces step-level supervision, which distributes signals across latent steps and improves stability. See extended discussion in Appendix B.

## 6 CONCLUSION

We introduce SIM-CoT, a training-based implicit reasoning method with step-level supervision on latent tokens. On GPT-2, SIM-CoT outperforms the strong explicit baseline SFT-CoT, while also surpassing implicit baselines such as Coconut and CODI. When scaling to larger LLaMA backbones, the performance achieves consistent gains over existing implicit reasoning methods and maintains fast inference efficiency. Ablation studies further show that it improves training stability with more latent tokens and can benefit from integration with training-free techniques such as soft thinking. Distance analysis confirms that SIM-CoT produces latent representations that are diverse yet stable.

## 7 ACKNOWLEDGEMENT

This project is funded in part by Shanghai Artificial Intelligence Laboratory, Shanghai Innovation Institute, the Centre for Perceptual and Interactive Intelligence (CPII) Ltd under the Innovation and Technology Commission (ITC)'s InnoHK. Dahua Lin is a PI of CPII under the InnoHK.

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

# APPENDIX

## USAGE OF LARGE LANGUAGE MODELS

In this paper, we used LLMs only for minor language polishing and formatting, without generating ideas, analyses, or experimental results.

## OUTLINE

In this appendix, we provide additional analyses and supporting materials to complement the main text. In Sec. A, we present experiments on combining soft thinking with SIM-CoT, including setup, results, and a detailed formulation with pseudocode. In Sec. B, we provide an overview of related work in explicit and implicit chain-of-thought reasoning. In Secs. C and D, we describe our implementation, hyperparameter configurations, boundary conditions for implicit token alignment, training overhead and training/inference procedures, including benchmark and dataset details. In Sec. E, we introduce the SIM-CoT training procedure and provide pseudocode for step-level supervision. In Sec. F, we offer geometric diagnostics of the latent space, analyzing inter-latent distances and distance to the vocabulary center. In Sec. G, we discuss interpretability analysis, including latent visualization and summary findings. Finally, we provide declarations on LLM usage and additional case studies on GSM8k to further illustrate the reasoning process and visualization choices.

## A ADDITIONAL ANALYSIS ON SOFT THINKING

Soft thinking (Zhang et al., 2025; Wu et al., 2025) is a training-free method for implicit reasoning in which the latent space is represented as a weighted average over the vocabulary embedding space. In contrast, SIM-CoT learns latent representations directly from data during training. To our knowledge, no prior work has evaluated a combination of these two approaches; our experiments provide the first such evaluation.

### A.1 SETUP

We apply the proposed soft thinking mechanism on top of both Coconut and SIM-CoT, while adopting GPT-2 as the backbone model. To assess the effectiveness of this approach, we perform evaluations on a diverse set of mathematical reasoning benchmarks. The in-domain evaluation is carried out on GSM8k-Aug, which provides augmented training and testing samples closely aligned with the original GSM8k distribution. To further examine generalization beyond the training domain, we include three out-of-domain benchmarks: GSM-Hard, which contains more challenging arithmetic problems with subtle variations in reasoning steps; MultiArith, which evaluates performance on multi-step arithmetic operations requiring careful sequencing of addition, subtraction, multiplication, and division; and SVAMP, which focuses on variations of elementary word problems designed to test robustness to superficial changes in problem statements.

### A.2 RESULTS

Table 5 (b) reports the results. Adding soft thinking improves accuracy in most cases. For Coconut, improvements are observed on GSM-Hard (+0.2) and MultiArith (+1.7), with a slight decrease on SVAMP (-0.2). For SIM-CoT, soft thinking consistently enhances performance: GSM8k-Aug (+0.2), GSM-Hard (+0.1), MultiArith (+0.7), and SVAMP (+0.1).

### A.3 FORMULATION

Let $z \in \mathbb{R}^d$ denote a continuous latent token, and $E \in \mathbb{R}^{|\mathcal{V}| \times d}$ be the embedding matrix of the vocabulary $\mathcal{V}$. Our goal is to enrich the representational capacity of $z$ by incorporating soft thinking, which allows the latent space to draw information not only from its continuous representation but also from the semantic structure of the vocabulary. The process can be described in three steps.

**Step 1. Vocabulary distribution.** The continuous latent token $z$ is first mapped into a probability distribution over the vocabulary space:

$$p = \text{softmax}(Wz),$$

where $W \in \mathbb{R}^{|\mathcal{V}| \times d}$ is the output projection matrix and $p \in \mathbb{R}^{|\mathcal{V}|}$ is the resulting distribution. This step can be viewed as interpreting the latent token in terms of vocabulary-level semantics, where each token in $\mathcal{V}$ is assigned a likelihood according to its relevance to $z$.

**Step 2. Soft-thinking embedding.** Using the distribution $p$, we compute a weighted mixture of vocabulary embeddings:

$$z_{\text{soft}} = E^\top p = \sum_{v \in \mathcal{V}} p_v \, E_v,$$

where $E_v$ is the embedding vector corresponding to token $v$. This operation can be seen as constructing a "soft token" that captures multiple semantic hypotheses simultaneously, instead of committing to a single discrete vocabulary token. As a result, $z_{\text{soft}}$ provides richer and smoother information than a hard token lookup.

**Step 3. Combination.** Finally, we combine the original continuous latent $z$ with the soft-thinking embedding $z_{\text{soft}}$:

$$z' = \alpha z + \beta z_{\text{soft}},$$

where $\alpha = \texttt{continuous\_weight}$ and $\beta = \texttt{soft\_weight}$ are hyperparameters that balance the contribution of the continuous and soft-thinking components. This formulation allows $z'$ to retain the model's learned continuous representations while also grounding them in the vocabulary space. Intuitively, the continuous part encourages compact reasoning within the latent space, whereas the soft-thinking component brings in semantic priors from the vocabulary, leading to more stable and interpretable reasoning.

The pseudocode implementation of the above process is presented as follows.

---

**Algorithm 1** Soft Thinking with Continuous Tokens

---

**Require:** Continuous latent $z$, embedding matrix $E$, weights $\alpha, \beta$
1: **if** $\beta > 0$ **then**
2:      Compute logits: $l \leftarrow Wz$
3:      Convert to probabilities: $p \leftarrow \text{softmax}(l)$
4:      Form soft embedding: $z_{\text{soft}} \leftarrow E^\top p$
5:      Update latent: $z' \leftarrow \alpha z + \beta z_{\text{soft}}$
6: **else**
7:      $z' \leftarrow z$
8: **end if**
9: **return** $z'$

---

### A.4 ANALYSIS

The results demonstrate that soft thinking complements training-based implicit reasoning. The hybrid latent $z'$ integrates semantics learned through training and distributional information from vocabulary mixing, which enables the model to explore diverse intermediate states rather than committing to a single deterministic path. This leads to improvements in both in-domain and out-of-domain benchmarks. Our findings suggest that combining training-free construction with training-based supervision provides gains beyond either approach in isolation.

## B RELATED WORK

**Explicit chain-of-thought reasoning.** Chain-of-thought (CoT) prompting enables large language models (LLMs) to generate intermediate reasoning steps before producing the final answer (Wei et al., 2022). This approach has been widely studied and extended in many directions. Self-consistency samples multiple reasoning paths and selects the majority answer to improve reliability (Wang et al.,

2023). Least-to-most prompting decomposes a complex question into simpler sub-problems and solves them in order (Zhou et al., 2023). Reflection-based reasoning allows the model to revise or verify its own intermediate steps, leading to better correctness (Shinn et al., 2023; Madaan et al., 2023). Other works focus on using external tools or symbolic solvers together with explicit reasoning, which further improves accuracy in mathematics and program synthesis (Yao et al., 2023). Methods such as progressive-hint prompting (Zheng et al., 2023) and step-level feedback (Wei et al., 2025) study how supervision can be incorporated into explicit reasoning to make reasoning more structured. Despite these advances, explicit CoT has clear drawbacks. Because it generates long token sequences, inference cost grows rapidly with reasoning length, and many intermediate steps are redundant or irrelevant to the final answer. Moreover, since explicit reasoning is restricted to tokens from a fixed vocabulary, it often commits to a single trajectory and shows limited reasoning diversity (Li et al., 2025; Zhang et al., 2025; Xu et al., 2025).

**Implicit chain-of-thought reasoning.** Implicit CoT performs multi-step computation in a continuous latent space instead of emitting long textual traces, reducing decoded length while keeping internal structure. Prior work follows four practical routes. First, **knowledge internalization** trains models to carry out reasoning internally by progressively removing explicit traces or by using dedicated control embeddings; examples include iCoT-SI (Deng et al., 2024), which removes steps during training to internalize reasoning. Second, **architectural modification** controls compute by reusing or skipping layers, or by adding light recurrence, so models can refine hidden states without lengthening outputs (Saunshi et al., 2025; Chen et al., 2025; Cheng & Van Durme, 2024; Su et al., 2025; Mohtashami et al., 2023; Geiping et al., 2025). Third, **training-free** methods construct continuous latents directly from the model's probability distribution over the vocabulary; Soft Thinking mixes embeddings by probability to form "concept" tokens that explore alternative paths without updating weights, which improves efficiency and diversity but does not bind each latent to step-level semantics (Zhang et al., 2025; Wu et al., 2025).

The fourth route, **auto-regressive latent reasoning**, updates and concatenates latent states in place of some token-level decoding and is the most relevant to our work (Xu et al., 2025; Tan et al., 2025). Coconut applies **answer-level** supervision—training on the final answer while leaving intermediate latents weakly constrained (Hao et al., 2025). CODI adds **trajectory-level** distillation by aligning an implicit trajectory with an explicit CoT trace, narrowing the gap to explicit CoT but giving only coarse guidance to intermediate steps (Shen et al., 2025). However, the implicit token length in CODI is fixed during training, which limits its flexibility and makes it less suitable for scaling to variable or longer reasoning chains. Our framework remains in the auto-regressive setting but changes the supervision: during training, each latent is aligned with its corresponding textual step (**step-level** supervision), distributing learning signals across the full latent chain to improve stability and semantic fidelity of intermediate states; at inference, the decoder is discarded, ensuring that the decoding cost remains identical to that of standard implicit CoT methods (e.g., Coconut).

## C  IMPLEMENTATION AND TRAINING DETAILS

We provide the full hyperparameter settings, training procedures, and additional analysis used in our experiments. Unless otherwise specified, we use the AdamW optimizer with $\beta_1 = 0.9$, $\beta_2 = 0.999$, and weight decay of $0.1$. Batch size is set to 128 for GPT-2 and LLaMA 1B, and 64 for LLaMA 3B and 8B. Early stopping is applied with a patience of 3 epochs. We now describe the training setups for Coconut, CODI, and our SIM-CoT.

### C.1  COCONUT TRAINING SETUP

Following Hao et al. (2025), GPT-2 and LLaMA 1B (Radford et al., 2019; Meta, 2024) are trained with a fixed learning rate of $1 \times 10^{-4}$. One implicit latent corresponds to two implicit tokens. A curriculum is applied: every three epochs, one explicit reasoning step is replaced by an implicit latent until the maximum number of latent steps is reached. After this expansion, training continues for 15 additional epochs.

## C.2 CODI TRAINING SETUP

For larger backbones such as LLaMA 3B and LLaMA 8B, we adopt task-specific hyperparameter settings to ensure stable training. In particular, we use a learning rate of $3 \times 10^{-4}$ for LLaMA 3B and train for 8 epochs, while for LLaMA 8B the learning rate is reduced to $1 \times 10^{-4}$ with 6 training epochs. These choices are motivated by the increased sensitivity of larger models to optimization dynamics, where smaller learning rates and fewer epochs help to prevent overfitting and instability.

When reproducing CODI on GPT-2 and LLaMA 1B, we strictly follow the configurations reported by Shen et al. (2025). Specifically, we use a learning rate of $3 \times 10^{-3}$ with 40 epochs for GPT-2, and a learning rate of $8 \times 10^{-4}$ with 10 epochs for LLaMA 1B. Adopting these settings ensures that our results are directly comparable to prior work and isolates the effect of our proposed method, rather than confounding it with differences in optimization schedules.

## C.3 SUMMARY OF HYPERPARAMETERS

Table 6: Training hyperparameters across different models.

| Model | Method | LR | Epochs |
|---|---|---|---|
| GPT-2 | Coconut | $1 \times 10^{-4}$ | 15 + curriculum |
| LLaMA 1B | Coconut | $1 \times 10^{-4}$ | 15 + curriculum |
| GPT-2 | CODI | $3 \times 10^{-3}$ | 40 |
| LLaMA 1B | CODI | $8 \times 10^{-4}$ | 10 |
| LLaMA 3B | CODI | $3 \times 10^{-4}$ | 8 |
| LLaMA 8B | CODI | $1 \times 10^{-4}$ | 6 |

## C.4 TRAINING-TIME OVERHEAD

We analyze the computational overhead introduced during training by the auxiliary decoder. Since the auxiliary decoder has the same architecture and number of parameters as the original decoder, and it participates in an additional forward pass during training, the overall parameter count and memory usage are approximately doubled compared with the implicit baselines Coconut and CODI.

To quantify the training-time overhead, Table 7 reports the wall-clock training hours of SIM-CoT and the corresponding implicit models under identical hardware settings (H800). Across different model scales, SIM-CoT introduces only a moderate increase in training time, ranging from approximately 2 hours for smaller backbones to around 16 hours for larger ones.

Table 7: Training hours of SIM-CoT compared with implicit baselines under identical hardware (H800).

| Model | Training Hours |
|---|---|
| Coconut GPT | ~180h |
| SIM-CoT (Coconut GPT) | ~192h |
| CODI GPT | ~16h |
| SIM-CoT (CODI GPT) | ~18.2h |
| CODI 1B | ~16.5h |
| SIM-CoT (CODI 1B) | ~18.5h |
| CODI 3B | ~34h |
| SIM-CoT (CODI 3B) | ~42h |
| CODI 8B | ~71h |
| SIM-CoT (CODI 8B) | ~87h |

These results show that the additional computational cost introduced by SIM-CoT remains modest relative to the improvements in stability and accuracy it provides.

## C.5 BOUNDARY CONDITIONS IN IMPLICIT TOKEN–STEP ALIGNMENT

Since the model produces a fixed number of implicit tokens ($K$) independent of the length of the ground-truth reasoning chain, we formalize the boundary conditions that govern how these tokens are aligned with textual reasoning steps. Let the annotated reasoning contain $N$ steps. Under this formulation, two boundary cases emerge:

**Case 1: $N > K$ (Longer reasoning chains).** When the reasoning chain contains more steps than latent tokens, a direct one-to-one alignment is impossible. We adopt a *many-to-one* strategy: the first $K - 1$ latent tokens are aligned individually to the first $K - 1$ reasoning steps, while the final token $z_K$ receives supervision from the concatenation of the remaining steps (steps $K$ through $N$). This boundary condition ensures that information from longer chains is preserved rather than truncated.

**Case 2: $N < K$ (Shorter reasoning chains).** When the reasoning chain is shorter than the number of latent tokens, the first $N$ latent tokens align one-to-one with the available reasoning steps. The remaining latent tokens ($z_{N+1}$ to $z_K$) are aligned with the final answer. This encourages the model to utilize its surplus latent capacity to refine the target solution.

These boundary rules guarantee that each implicit token is assigned a semantically meaningful supervision signal, regardless of how the reasoning length compares with the fixed latent budget $K$. They also provide stability during training by preventing both supervision sparsity (when $N < K$) and information loss (when $N > K$).

# D TRAINING AND INFERENCE DETAILS

**Curriculum for $K$.** We use a curriculum schedule to gradually increase the number of implicit steps. Each latent corresponds to two implicit tokens. Let $K_{\max}$ denote the maximum number of latents. Starting from $K^{(0)} = 0$, the number of implicit steps after epoch $e$ is

$$K^{(e)} \;=\; \min\left(K_{\max}, \left\lfloor \tfrac{e}{\Delta e} \right\rfloor\right),$$

where $\Delta e$ is the update interval in epochs. Once $K^{(e)}$ reaches $K_{\max}$, it remains fixed for the remainder of training.

**Inference and Efficiency.** At inference time, the auxiliary decoder is removed and only the base model is executed:

$$U^{(0)} = [\, e(x_1), \ldots, e(x_T)\,], \quad \text{for } k = 1, \ldots, K: \ z_k = H_\theta(U^{(k-1)}), \quad U^{(k)} = U^{(k-1)} \oplus z_k,$$

and after $K$ implicit steps the model switches back to explicit decoding to generate the final answer sequence $a$ as in Eq. equation 4. The total decoding length is $T + K + L_a$, where $T$ is the input length, $K$ the number of implicit steps, and $L_a$ the answer length. In practice, the cost is comparable to other implicit reasoning methods because $K$ is moderate. In tasks where explicit CoT requires long trajectories ($L_{\mathrm{CoT}} \gg K$), the implicit formulation reduces decoding positions, providing efficiency gains without loss of reasoning accuracy.

**Benchmark Detail.** **SVAMP** (Patel et al., 2021) (Simple Variations on Arithmetic Math Word Problems) is a benchmark dataset designed to test the robustness of math word problem solvers to superficial changes. It contains 1,000 elementary-level arithmetic word problems (grade 4 and below), each involving a single unknown and solvable by an arithmetic expression with no more than two operators. The problems are transformations of existing datasets (such as MAWPS and ASDiv-A) with controlled variations in wording, structure, and number values to reduce artifacts and superficial cues. SVAMP's average number of reasoning steps required is around 1.2, similar to the base datasets, but model performance drops significantly when tested on SVAMP, showing that many models rely on heuristic patterns rather than deep understanding.

**GSM-Hard** (Gao et al., 2022) is a more challenging variant of the GSM8K dataset, intended to test models' ability to cope with harder numerical values. It retains the same problem statements as the original GSM8K, but replaces many of the numbers with larger and less common numerical

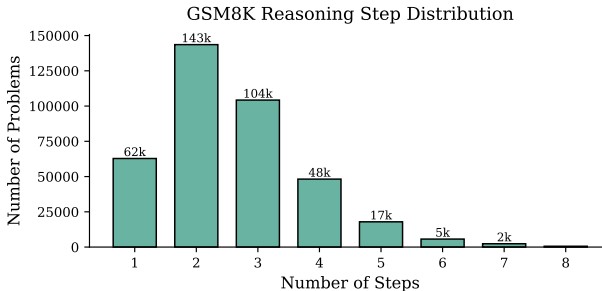

Figure 5: Distribution of reasoning steps in the GSM8K-Aug training dataset. Most problems involve two to four steps, with a long-tail of harder cases. For visualization, step counts with fewer than 200 problems are omitted, though all examples are used in training.

values, making superficial arithmetic computation and reasoning harder. The dataset has about 1,319 examples (matching the GSM8K test set size).

**MultiArith** (Roy & Roth, 2015) is a dataset of multi-step arithmetic word problems designed to challenge systems to correctly sequence multiple operations. It contains 600 problems collected from educational sources, each solvable by one equation involving two or more of the four basic operations (addition, subtraction, multiplication, division). The problems require reasoning over multiple sentences to extract and combine numeric quantities, understand the implied operations, and compute the result. MultiArith has been widely used as a benchmark for evaluating arithmetic reasoning generalization, particularly for models that go beyond single-operation problems. Analyses show that many models struggle on these examples compared to simpler datasets, highlighting the importance of handling compositional and sequential numerical reasoning.

**Training Data.** GSM8K-Aug (Deng et al., 2024) is the only training corpus we use. It is an augmented dataset derived from GSM8K (Cobbe et al., 2021), expanding the original 8.5k training problems to roughly 385k examples through paraphrasing, numerical resampling, and synthetic generation with GPT-4. The distribution of reasoning steps in GSM8K-Aug is illustrated in Fig. 5, where the majority of problems require two to four steps, while a long-tail of six or more steps persists. This balance of common and complex instances makes GSM8K-Aug particularly suitable for training models that need to generalize across reasoning difficulty levels.

## E    SIM-CoT Training Implementation

We provide pseudocode for the SIM-CoT training process, which illustrates how continuous latent embeddings are aligned with explicit supervision at the step level. In particular, each reasoning step in the explicit chain is mapped to a corresponding latent representation, and the training objective enforces consistency between the predicted latent tokens and the ground-truth step annotations. This design ensures that the model learns to represent intermediate reasoning steps in a compact latent space while still retaining interpretability through explicit alignment. By supervising at the step level rather than only at the final answer or trajectory level, SIM-CoT enables finer control over the reasoning process and reduces instability that often arises when scaling to longer chains or larger numbers of implicit tokens.

## F    Geometric Diagnostics of the Latent Space

We analyze the geometry of latent representations with two metrics.

*Inter-latent distance.*

$$\text{Dist}(z_{1:K}) = \frac{2}{K(K-1)} \sum_{1 \le i < j \le K} \left\| z_i - z_j \right\|_2. \tag{9}$$

A larger value indicates better separation, reducing the risk of collapse.

*Distance to vocabulary center.* Let $\mu = \frac{1}{|\mathcal{V}|} \sum_{v \in \mathcal{V}} E_v$ denote the mean embedding. Then,

$$\text{DistVC}(z_{1:K}) = \frac{1}{K} \sum_{k=1}^{K} \left\| z_k - \mu \right\|_2. \tag{10}$$

---

**Algorithm 2** SIM-CoT Training Procedure

---

**Require:** Batch size $b$, number of thoughts $C$, continuous embeddings $Z$, tokenized inputs $X$, embedding matrix $E$
 1: **for** each thought $t = 1, \ldots, C$ **do**
 2:     **for** each sample $i = 1, \ldots, b$ **do**
 3:         Extract continuous embeddings $z_{i,t}$ from $Z$
 4:         Obtain token embeddings $e_{i,t}$ from $E(X_{i,t})$
 5:         Concatenate embeddings: $h_{i,t} \leftarrow [z_{i,t}; e_{i,t}]$
 6:         Build attention mask $m_{i,t}$ up to EOS
 7:         Assign position ids $p_{i,t}$
 8:         Prepare labels $y_{i,t}$ with masked tokens set to $-100$
 9:     **end for**
10: **end for**
11: Pad and stack $\{h, m, p, y\}$ to maximum sequence length
12: Prepare 4D attention mask: $\hat{M} \leftarrow \text{PrepareMask}(M)$
13: Forward pass: $\hat{O} \leftarrow \text{ExplainableLLM}(H, \hat{M}, P)$
14: Extract logits: $L \leftarrow \hat{O}.\text{logits}$
15: Shift logits and labels: $L' \leftarrow L[:, :-1]$, $Y' \leftarrow Y[:, 1:]$
16: Compute cross-entropy loss: $\ell = \text{CrossEntropy}(L', Y')$
17: Normalize $\ell$ over valid positions
**Ensure:** Final training loss $\ell$

---

Moderate values indicate that latents remain close enough to the lexical manifold for stability, while avoiding collapse toward the center. These diagnostics are not used in training but serve as indicators of diversity and stability in the learned latent space.

## G   ADDITIONAL DETAILS FOR INTERPRETABILITY ANALYSIS

### G.1   MAKING IMPLICIT REASONING VISIBLE

Continuous thoughts produced by implicit reasoning models are represented as latent embeddings that do not correspond to discrete vocabulary tokens, and therefore cannot be directly decoded by a tokenizer. This makes it difficult to interpret how the model internally organizes multi-step reasoning. To address this, we reuse the decoder that was employed for step-level supervision during training, and apply it at inference time to map each latent embedding into a human-readable token sequence.

As illustrated in Figure 4, the process begins with a natural language problem (e.g., a math word problem) that is embedded and passed into the large language model. The model generates a sequence of implicit latent tokens, which capture intermediate reasoning steps in continuous space. These latent tokens are then fed into the optional decoder, which translates them into interpretable expressions. Each latent corresponds to one reasoning step, and the autoregressive generation order encodes the dependency structure across steps.

For example, in the GSM8k case study shown in the figure, the first latent is decoded as $0.3 \times 120 = 36$, representing the number of watermelons harvested initially. The second latent builds upon this result to compute $120 - 36 = 84$, the remaining melons. The third latent then calculates $\frac{3}{4} \times 84 = 63$, and the final latent derives the answer $84 - 63 = 21$. For clarity and to save space, the figure merges the second and third steps into a single box, but the actual implicit reasoning unfolds across four distinct latent steps. This sequence of decoded latents mirrors the logic of explicit chain-of-thought reasoning, while being produced implicitly within the latent space.

By projecting implicit tokens into interpretable space, we gain direct visibility into how the model structures multi-step reasoning. This not only enables analysis of the correctness and consistency of intermediate steps, but also highlights the dependencies across steps that underlie the final prediction. The visualization confirms that SIM-CoT can encode semantically meaningful and logically ordered reasoning steps in its latent space, bridging the gap between implicit and explicit reasoning.

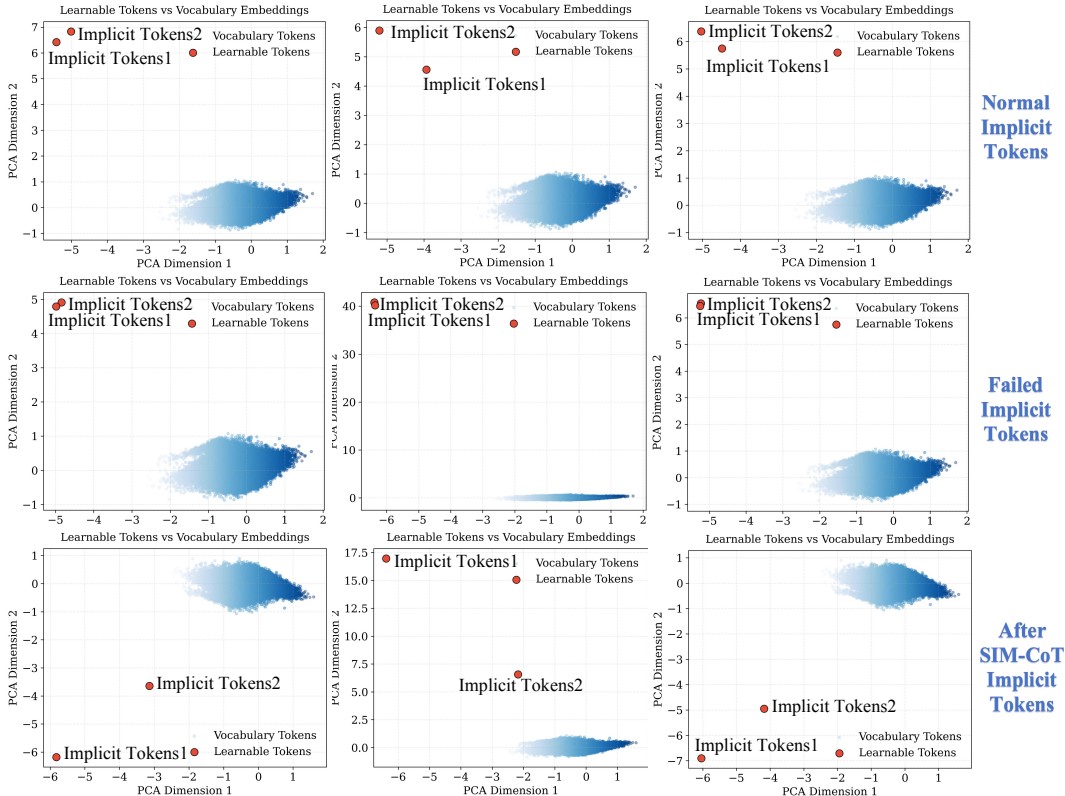

Figure 6: Visualization of distances among implicit tokens and their distances to the vocabulary center. The first row shows normal implicit tokens with well-separated representations, the second row illustrates failed implicit tokens where distances collapse and drift away from the vocabulary center, and the third row presents implicit tokens after applying SIM-CoT, which restores both separation and stability in the latent space.

## G.2 SUMMARY

Overall, the results demonstrate that SIM-CoT establishes a balance between **diversity** and **stability** in the latent space. Larger inter-latent distances mitigate representation collapse, while moderate distances to the vocabulary center prevent excessive drift. This equilibrium supports stable implicit reasoning and provides robustness when scaling to more latent tokens.

## H ADDITIONAL CASE STUDIES ON GSM8K

In practice, implicit reasoning continues to produce latent tokens even after the correct answer has been reached. These trailing latents no longer introduce new steps but simply repeat the final prediction. For clarity, we omit such redundant tokens in the visualizations. As a result, only the latents that correspond to meaningful intermediate steps are displayed in Figure 7, while those mapping directly to the final answer are hidden. This choice improves readability without changing the underlying reasoning process.

Notably, the decoded reasoning steps consistently match the semantic structure of explicit chain-of-thought annotations, while being generated implicitly within the latent space. The final predictions align with the ground-truth answers, demonstrating that SIM-CoT is capable of encoding interpretable and step-ordered reasoning without requiring explicit supervision at inference time.

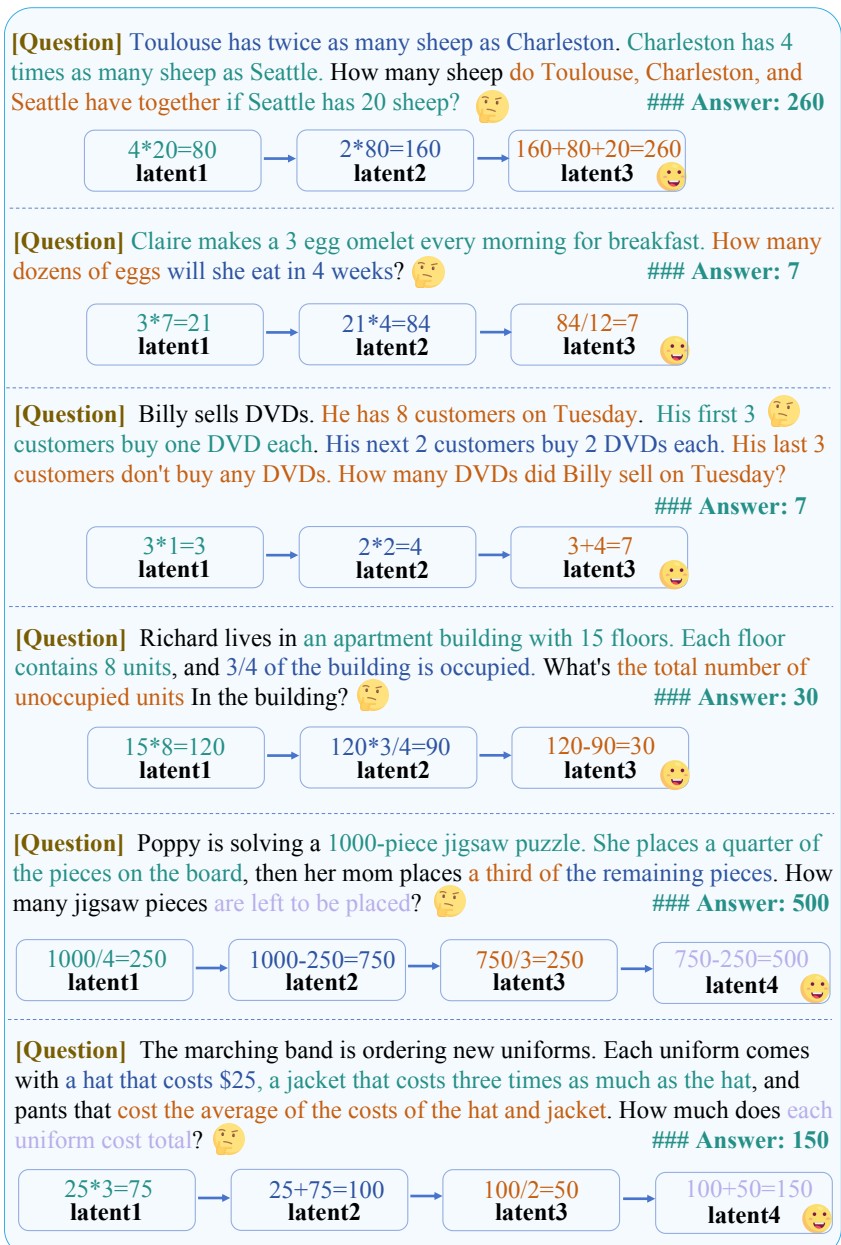

Figure 7: Additional SIM-CoT case studies on GSM8k. Each example illustrates how implicit latent tokens correspond to intermediate reasoning steps. Arrows indicate the dependency relations across steps, while colored spans in the question highlight the textual evidence that supports each step. The decoded sequence of latent steps produces the correct final answer, which matches the ground-truth label.

