# OpenReview forum: "SIM-CoT: Supervised Implicit Chain-of-Thought"
_ICLR.cc/2026/Conference — ICLR 2026 Poster_

### Official Review · Reviewer_URA3 · 2025-10-16

**Soundness:** 4
**Presentation:** 3
**Contribution:** 2
**Rating:** 6
**Confidence:** 5

**Summary:**

The paper introduces SIM-CoT, a training module that improves implicit Chain-of-Thought (CoT) reasoning in Large Language Models (LLMs). It addresses the performance gap and instability issues found in current implicit CoT methods.

Implicit CoT methods suffer from a "latent instability" issue when their computational budget (number of reasoning tokens) is increased. The authors' analysis shows this instability arises because latent representations become homogeneous and lose semantic diversity, which is caused by insufficient step-level supervision. This can cause training to collapse and performance to drop sharply.

During training, it uses an auxiliary decoder to align each implicit token with its corresponding explicit reasoning step. This ensures the latent states capture distinct information. The auxiliary decoder is removed during inference, preserving the token efficiency of implicit CoT with no added overhead. The decoder allows for the visualization and diagnosis of each latent reasoning step.

It provides a systematic analysis of the latent instability issue in implicit CoT approaches, showing it stems from insufficient supervision.
The introduction of SIM-CoT, which uses step-level supervision to improve performance and stability with minimal inference overhead.
Extensive experimental validation showing consistent performance gains across a range of LLMs, including GPT-2 and LLaMA 3 models.

**Strengths:**

The primary originality lies in the identification and framing of the "latent instability issue". While the performance gap between implicit and explicit Chain-of-Thought (CoT) was known, this paper is the first to provide a rigorous, multi-faceted diagnosis of why scaling implicit methods fails. It introduces novel analytical concepts like "semantic homogenization" to explain the collapse of the latent space, a significant contribution to understanding the failure modes of these models.
The proposed SIM-CoT method is a creative and effective combination of existing architectural components. It leverages an auxiliary decoder—a standard component—in a novel way: as a temporary, training-only tool for applying fine-grained, step-level supervision to otherwise unconstrained latent states. This is a departure from prior work that relied on coarser answer-level or trajectory-level supervision.
The work directly addresses and removes a critical bottleneck in implicit CoT research: the inability to effectively scale the computational budget (i.e., the number of latent tokens). By stabilizing the training process, SIM-CoT allows for the use of more latent tokens, which in turn boosts performance where previous methods would fail.
The central claims are exceptionally well-supported. The diagnosis in Figure 1 is not based on a single metric but is corroborated through a comprehensive analysis of accuracy, information loss (operator vs. number), geometric properties of the latent space, and qualitative examples of decoded latents.
The experimental setup is of high quality. The authors compare against a comprehensive suite of strong baselines, including explicit CoT and state-of-the-art implicit methods like CODI. The evaluation is robust, conducted across multiple model families (GPT-2, LLaMA) and sizes (1B to 8B) on both in-domain and out-of-domain benchmarks to ensure the findings are generalizable.

The paper follows a clear, logical narrative that is easy to follow. It begins by identifying a problem, provides a detailed diagnosis, proposes a well-motivated solution, and validates it with extensive experiments. The writing is precise and unambiguous.
This work marks a significant milestone for implicit reasoning. It is the first training-based approach to show that an implicit CoT method can surpass a strong explicit CoT baseline on a standard benchmark, both in accuracy and token efficiency. This result helps to close a critical performance gap and makes implicit reasoning a more viable and attractive alternative for practical applications.
SIM-CoT is designed as a "plug-and-play" module that is shown to boost the performance of multiple existing implicit methods. This high degree of applicability makes the contribution immediately useful to other researchers. The gains in inference efficiency are substantial and directly address one of the most pressing challenges in deploying large-scale reasoning models

**Weaknesses:**

1. The method enforces a strict one-to-one mapping between implicit tokens and textual reasoning steps, which may not reflect the variable complexity of real-world reasoning.
Suggestion: Explore flexible alignment mechanisms, such as allowing a variable number of latent tokens per reasoning step based on its complexity.

2. At inference, the model generates a fixed number of implicit tokens (K), which is inefficient as problems require different reasoning lengths.
Suggestion: Implement a dynamic stopping mechanism, such as training the model to generate a special "end-of-thought" latent token to terminate the reasoning phase adaptively.

3. The paper notes that using larger decoders degrades performance but offers only a speculative explanation.
Suggestion: Conduct a deeper analysis to explain this counterintuitive result, for example, by measuring gradient flows or using representation analysis to check for "misalignment" between the LLM and decoder.

4. The paper highlights the lack of inference overhead but does not discuss the increased computational cost (memory, time) during training due to the auxiliary decoder.
Suggestion: Quantify the training-time overhead (e.g., increase in parameters, memory usage, and wall-clock time) to provide a complete picture of the method's trade-offs.

5. The evaluation is confined to mathematical reasoning benchmarks. Claims about improving "complex reasoning" would be stronger if validated on a broader range of tasks.
Suggestion: Evaluate the method on other domains, such as commonsense (e.g., StrategyQA) or symbolic reasoning, to demonstrate the generalizability of the approach beyond arithmetic problems.

**Questions:**

This paper introduces SIM-CoT, a novel training module that addresses a key limitation in implicit Chain-of-Thought (CoT) reasoning. The authors identify and provide a rigorous diagnosis of a "latent instability issue" that causes performance to collapse when scaling implicit methods. The proposed solution, which uses a training-only auxiliary decoder to apply step-level supervision, is both elegant and effective.

The paper's strengths are significant. The diagnosis of the problem is original and well-supported by a multi-faceted analysis. The experimental results are strong and comprehensive, demonstrating that SIM-CoT not only boosts the performance of existing state-of-the-art implicit methods across various model scales but can also surpass a strong explicit CoT baseline in both accuracy and token efficiency. The clarity of the presentation, particularly the high-quality figures and diagrams, makes the work easy to understand and appreciate. The contribution is highly significant, offering a practical path toward making efficient reasoning methods more powerful and reliable.

The paper has a few limitations that should be addressed. The current method relies on a rigid, one-to-one alignment between latent tokens and reasoning steps and uses a fixed number of reasoning steps (K) at inference, which may not be optimal for problems of varying complexity. The authors also do not discuss the training-time overhead introduced by the auxiliary decoder, focusing only on inference efficiency. Finally, the experimental validation is confined to mathematical reasoning tasks, which limits the claims about general "complex reasoning".

---

> ### Author Response · Authors · 2025-11-21
> **Response to Reviewer URA3 (1/3)**
>
> Dear Reviewer URA3,
>
> Thanks for your valuable feedback. We sincerely thank all reviewers for acknowledging that this paper provides a novel and rigorous diagnosis of latent instability (4kCa, URA3), introduces a well-motivated step-level supervision framework (4kCa, UZJG, URA3), achieves significant improvements over explicit and implicit CoT with full inference efficiency (4kCa, 7Zpg, URA3), offers interpretability through per-step latent visualization (4kCa), presents a plug-and-play design with broad applicability (7Zpg, URA3), and is clearly written and easy to follow (7Zpg, URA3). Below we address your questions:
>
> > Q1: The method enforces a strict one-to-one mapping between implicit tokens and textual reasoning steps, which may not reflect the variable complexity of real-world reasoning. Suggestion: Explore flexible alignment mechanisms, such as allowing a variable number of latent tokens per reasoning step based on its complexity.
>
> Thank you for this thoughtful suggestion. Our paper focuses on a specific goal: introducing **step-level implicit supervision** to address the latent instability problem in prior implicit CoT methods. The strict one-to-one alignment between implicit tokens and textual reasoning steps is a design choice that supports this contribution by providing stable supervision signals during training.
>
> We appreciate your idea of exploring **flexible alignment mechanisms**, such as allowing a variable number of latent tokens per step based on step complexity. This is an interesting direction and could potentially improve adaptability in real-world reasoning scenarios. However, such a mechanism extends beyond the scope of the current work, which aims to establish a clear and controlled supervision scheme to diagnose and resolve instability in the latent space.
>
> We will consider this idea in future work, where more dynamic alignment strategies—possibly combined with reinforcement learning for adaptive supervision—could complement the framework proposed in this paper.
>
> > Q2: At inference, the model generates a fixed number of implicit tokens (K), which is inefficient as problems require different reasoning lengths. Suggestion: Implement a dynamic stopping mechanism, such as training the model to generate a special "end-of-thought" latent token to terminate the reasoning phase adaptively.
>
> Thank you for raising this point and for suggesting the idea of using a dynamic stopping mechanism, such as an “end-of-thought” latent token. Our work focuses on a specific contribution: introducing **step-level implicit supervision** to address latent instability and improve the semantic diversity of implicit tokens. To clearly isolate and study this effect, our design adopts a fixed number of implicit tokens during inference.
>
> We agree that an adaptive mechanism for determining the number of implicit tokens is an interesting direction, especially since real problems vary in reasoning depth. However, exploring dynamic stopping policies goes beyond the scope of the present work, whose primary goal is to analyze instability and demonstrate that step-level supervision can stabilize and enrich the latent reasoning space. Future work could incorporate your suggestion, and approaches such as learning an end-of-thought signal or using reinforcement learning to adaptively terminate reasoning are promising extensions that we plan to investigate. We appreciate your constructive suggestion.
>
> > Q3: The paper notes that using larger decoders degrades performance but offers only a speculative explanation. Suggestion: Conduct a deeper analysis to explain this counterintuitive result, for example, by measuring gradient flows or using representation analysis to check for "misalignment" between the LLM and decoder.
>
> Thank you for the helpful suggestion. The performance drop observed when using larger decoders indeed calls for deeper analysis. Since this direction is not the central focus of our study, the initial version does not examine it in detail. Our main contribution is to show that step-level implicit supervision improves the stability and semantic diversity of latent tokens, and scaling the decoder introduces substantial training cost without directly advancing this goal.
>
> Following your suggestion, we quantify encoder–decoder compatibility (1B encoder vs. various decoders) via averaged Euclidean distance of last hidden states on GSM8K; results follow.
>
> |Encoder–Decoder Pair|Euclidean Distance (GSM8K avg.)|
> |-|-|
> |1B encoder – 1B decoder|0.007|
> |1B encoder – 3B decoder|0.14|
> |1B encoder – 8B decoder|0.38|
>
> The trend shows that larger decoders diverge more strongly from the encoder representation space, which supports the view that increasing decoder size introduces representation misalignment and contributes to the observed degradation in performance. This observation aligns with the empirical findings in Table 4(a) of the original paper, where the 1B decoder achieves the best results.

---

> > ### Author Response · Authors · 2025-11-21
> > **Response to Reviewer URA3 (2/3)**
> >
> > > Q4: The paper highlights the lack of inference overhead but does not discuss the increased computational cost (memory, time) during training due to the auxiliary decoder. Suggestion: Quantify the training-time overhead (e.g., increase in parameters, memory usage, and wall-clock time) to provide a complete picture of the method's trade-offs.
> >
> > Thank you for raising the concern regarding the training cost of our method. In terms of parameter count and memory usage, the introduced decoder has the same architecture and the same number of parameters as the original model, and it also requires a forward pass during training. As a result, both the parameter count and memory usage are roughly **twice** those of the original implicit baselines (Coconut and CODI).
> >
> > To quantify the training-time overhead, we report the training hours of SIM-CoT compared with the corresponding implicit models (Coconut and CODI) on the same hardware (H800 hours):
> >
> > | Model                 | Training Hours |
> > |-----------------------|----------------|
> > | Coconut GPT           | ~180h          |
> > | SIM-CoT (Coconut GPT) | ~192h          |
> > | CODI GPT              | ~16h           |
> > | SIM-CoT (CODI GPT)    | ~18.2h         |
> > | CODI 1B               | ~16.5h         |
> > | SIM-CoT (CODI 1B)     | ~18.5h         |
> > | CODI 3B               | ~34h           |
> > | SIM-CoT (CODI 3B)     | ~42h           |
> > | CODI 8B               | ~71h           |
> > | SIM-CoT (CODI 8B)     | ~87h           |
> >
> > As shown above, the additional training time brought by SIM-CoT remains modest across different model scales, ranging from an increase of about 2 hours for smaller models (such as CODI GPT) to about 16 hours for larger models (such as CODI 8B). We view this increase as a reasonable trade-off in exchange for the performance gains achieved through the supervised decoder.
> >
> > We appreciate your suggestion and add a discussion of the training overhead to  Appendix C.4(highlighted in red in the manuscript).
> >
> > > Q5: The evaluation is confined to mathematical reasoning benchmarks. Claims about improving "complex reasoning" would be stronger if validated on a broader range of tasks. Suggestion: Evaluate the method on other domains, such as commonsense (e.g., StrategyQA) or symbolic reasoning, to demonstrate the generalizability of the approach beyond arithmetic problems.
> >
> > Thank you for your feedback. We understand the concern that the evaluation has primarily focused on math-style reasoning, potentially leaving generalization to other domains less explored. In response to this, we have expanded our evaluation to include a broader range of reasoning tasks, such as CommonsenseQA and StrategyQA, which test the generalization ability of our method, SIM-CoT, across different domains.
> >
> > The results, as shown in the table below, demonstrate SIM-CoT's strong performance across both CommonsenseQA and StrategyQA, which are domains beyond math-style reasoning. For example, on CommonsenseQA, SIM-CoT with GPT-2 and LLaMA 1B models **outperforms explicit reasoning baseline** SFT-CoT, **as well as implicit reasoning baselines** such as Coconut and CODI, achieving scores of **46.2 and 74.6**, respectively. Similarly, on StrategyQA, SIM-CoT achieves notable results, with scores of **56.3 and 65.3** for GPT2 and LLaMA 1B, respectively.
> >
> > |          |         | CommonsenseQA |      |         |
> > |----------|---------|---------------|------|---------|
> > | Backbone | SFT-CoT | Coconut       | CODI | SIM-CoT |
> > | GPT-2    | 36.9    | 38.6          | 44.1 | **46.2**|
> > | LLaMA 1B | 67.9    | 60.6          | 73.8 | **74.6**|
> >
> > |          |         | StrategyQA |      |         |
> > |----------|---------|------------|------|---------|
> > | Backbone | SFT-CoT | Coconut    | CODI | SIM-CoT |
> > | GPT-2    | 38.1    | 42.6       | 52.1 | **56.3**|
> > | LLaMA 1B | 39.7    | 43.2       | 63.8 | **65.3**|
> >
> > These results provide evidence that SIM-CoT generalizes well to various domains, demonstrating its ability to handle diverse reasoning tasks beyond math-style problems. We believe this expanded evaluation addresses the concern raised and highlights the versatility of our approach.
> >
> > We hope these additional results help clarify the generalization abilities of our method, and we welcome any further suggestions on how to enhance the evaluation process.

---

> > > ### Author Response · Authors · 2025-11-21
> > > **Response to Reviewer URA3 (3/3)**
> > >
> > > > Q6: The paper has a few limitations that should be addressed. The current method relies on a rigid, one-to-one alignment between latent tokens and reasoning steps and uses a fixed number of reasoning steps (K) at inference, which may not be optimal for problems of varying complexity. The authors also do not discuss the training-time overhead introduced by the auxiliary decoder, focusing only on inference efficiency. Finally, the experimental validation is confined to mathematical reasoning tasks, which limits the claims about general "complex reasoning".
> > >
> > > Thank you for your thoughtful comments. We appreciate the opportunity to clarify these points.
> > >
> > > Regarding the one-to-one alignment and the fixed number of latent tokens (K), these choices are deliberate design decisions aimed at isolating and studying our core contribution—**providing stable step-level implicit supervision to address latent instability in prior implicit CoT approaches**. More flexible mechanisms (e.g., variable token–step alignment or dynamic stopping policies) are indeed promising, but they extend beyond the scope of this work. As noted in our previous responses, we view such mechanisms—possibly through end-of-thought signals or RL-based adaptive control—as valuable directions for future research.
> > >
> > > On training-time overhead, we now include a detailed quantification. The auxiliary decoder introduces additional parameters and a forward pass, roughly doubling parameter count and memory usage compared with implicit baselines. Empirically, the added wall-clock cost is modest: SIM-CoT increases training time by **~2 hours for small models** and up to **~16 hours for larger models**, which we believe is a reasonable trade-off for the stability and performance gains it brings.
> > >
> > > Finally, we have expanded the evaluation beyond mathematical reasoning. New experiments on CommonsenseQA and StrategyQA demonstrate that **SIM-CoT generalizes effectively across domains**, outperforming both explicit and implicit baselines. These results strengthen the claim that our method benefits general complex reasoning rather than only arithmetic tasks.
> > >
> > > We hope this clarifies our design choices and addresses the concerns raised.

---

> ### Author Response · Authors · 2025-11-27
>
> Dear Reviewer URA3,
>
> Thank you very much for your detailed comments and helpful suggestions on our paper.
>
> We would like to kindly inquire if our rebuttal has addressed your concerns adequately. If there is anything else we can clarify or elaborate on to support your assessment, we would be more than willing to provide additional information.
>
> We greatly appreciate your time and contribution.
>
> Best regards,
>
> Authors

---

### Official Review · Reviewer_7Zpg · 2025-10-28

**Soundness:** 3
**Presentation:** 3
**Contribution:** 2
**Rating:** 6
**Confidence:** 4

**Summary:**

The paper introduces SIM-CoT that improves implicit reasoning in LLMs. Unlike prior implicit CoT methods such as Coconut and CODI, which only provide coarse supervision at the answer or trajectory level, SIM-CoT introduces fine-grained, step-level supervision. Empirical results show that SIM-CoT is a plug-and-play module that can be seamlessly combined with existing implicit reasoning methods to enhance both accuracy and stability.

**Strengths:**

1. The proposed method is useful, especially since it functions as a plug-and-play module that can be easily combined with other approaches.
2. The paper’s presentation is overall clear and well-structured, with a logical and coherent flow of ideas.

**Weaknesses:**

1. At the beginning, the paper claims that one of the reasons for the instability of prior implicit CoT methods is that implicit tokens **collapse into homogeneous latent states**. This part could be better supported experimentally. Although the authors show that their method is indeed more stable than Coconut when the number of latent tokens increases (figure 3), it would be even more convincing if they included additional measurements or visualizations of the heterogeneity of latent tokens (for example, some measure related to the distance of latent tokens) in the main body.

2. While the proposed method introduces more fine-grained step-level supervision compared to prior approaches, this also adds some trouble to training: the model must rely on **preprocessed, structured**, fine-grained training data, which may limit the scalability of the method.

3. The authors mention that the extra decoder model is architecturally identical to the LLM (line 252). This means the two share parameters, or if the decoder is a smaller, separate model?

4. Although the authors report the average number of tokens used during inference, the introduced decoder model also generates a sequence of tokens at each step for supervision. My concern is that the training cost might be higher than Coconut or even SFT-CoT. A more detailed discussion of training overhead would help clarify the efficiency and practicality of the proposed method.

5. Sections 3.1–3.5 are presented clearly but the notations are quite dense. It would be more reader-friendly if some of the notations were also depicted in Figure 2, which would help distinguish the different symbols more easily.

**Questions:**

see Weakness.

---

> ### Author Response · Authors · 2025-11-21
> **Response to Reviewer 7Zpg (1/3)**
>
> Dear Reviewer 7Zpg,
>
> Thanks for your valuable feedback. We sincerely thank all reviewers for acknowledging that this paper provides a novel and rigorous diagnosis of latent instability (4kCa, URA3), introduces a well-motivated step-level supervision framework (4kCa, UZJG, URA3), achieves significant improvements over explicit and implicit CoT with full inference efficiency (4kCa, 7Zpg, URA3), offers interpretability through per-step latent visualization (4kCa), presents a plug-and-play design with broad applicability (7Zpg, URA3), and is clearly written and easy to follow (7Zpg, URA3). Below we address your questions:
>
> > Q1: At the beginning, the paper claims that one of the reasons for the instability of prior implicit CoT methods is that implicit tokens collapse into homogeneous latent states. This part could be better supported experimentally. Although the authors show that their method is indeed more stable than Coconut when the number of latent tokens increases (figure 3), it would be even more convincing if they included additional measurements or visualizations of the heterogeneity of latent tokens (for example, some measure related to the distance of latent tokens) in the main body.
>
> Thank you for this constructive suggestion. In the original version, we place part of the heterogeneity analysis in the Interpretability of Implicit Reasoning ablation section in the main body, with the full results included in Appendix I due to page limits. This arrangement reflects our intent to first present the stabilizing effect of SIM-CoT and then connect it back to the analysis discussed in the introduction.
>
> Following your advice, we now move all of this analysis (including the table) into the main body of the paper. (highlighted in red in the experiment section). The updated section includes distance-based measurements that quantify the heterogeneity of latent tokens and demonstrate that SIM-CoT effectively mitigates latent homogenization and promotes a more diverse latent space. This additional evidence directly strengthens the empirical support for our initial claim.

---

> > ### Author Response · Authors · 2025-11-21
> > **Response to Reviewer 7Zpg (2/3)**
> >
> > > Q2: While the proposed method introduces more fine-grained step-level supervision compared to prior approaches, this also adds some trouble to training: the model must rely on preprocessed, structured, fine-grained training data, which may limit the scalability of the method.
> >
> > We thank the reviewer for this question regarding generalizability. Our approach uses structured reasoning steps, which have been shown in many studies to support higher reasoning accuracy and reduce redundant reasoning paths [1, 2, 3]. While our primary experiments focused on datasets where process-level reasoning annotations are readily available, our method is **not** limited to them. For datasets lacking explicit process-level annotations, we propose a practical two-stage approach:
> >
> > (1) **Generate Structured Traces**: First, we use a powerful external model to generate structured reasoning paths. This aligns with recent work showing that models like DeepSeek-R1 [4] are highly effective at converting unstructured text into structured reasoning chains [4, 5, 6]. (2) **Apply SIM-CoT**: With these generated traces, we then apply our SIM-CoT training, using the structured steps as the supervision signal.
> >
> > To validate this approach, we conduct new experiments on two non-mathematical reasoning benchmarks, CommonsenseQA [7] and StrategyQA [8]. We use the open-source model DeepSeek-R1-Distill-Qwen-7B [4] to generate the reasoning steps and then apply our training procedure. The results below show that SIM-CoT consistently and significantly outperforms all baselines, demonstrating its robust generalizability.
> >
> > |||CommonsenseQA|||
> > |-|-|-|-|-|
> > |Backbone|SFT-CoT|Coconut|CODI|SIM-CoT|
> > |GPT-2|36.9|38.6|44.1|**46.2**|
> > |LLaMA 1B|67.9|60.6|73.8|**74.6**|
> >
> > |||StrategyQA|||
> > |-|-|-|-|-|
> > |Backbone|SFT-CoT|Coconut|CODI|SIM-CoT|
> > |GPT-2|38.1|42.6|52.1|**56.3**|
> > |LLaMA 1B|39.7|43.2|63.8|**65.3**|
> >
> > As the tables show, SIM-CoT achieves the highest scores across both datasets and model sizes. On CommonsenseQA, it surpasses the explicit SFT-CoT baseline by up to **9.3 points**. The gains are even more pronounced on the more complex StrategyQA dataset, where SIM-CoT outperforms SFT-CoT by a remarkable **25.6 points** with the LLaMA 1B model. These findings confirm that our proposed pipeline effectively addresses datasets without pre-existing reasoning annotations and that SIM-CoT is a highly effective method for improving reasoning performance in these more general domains.
> >
> > [1] Shunyu Yao, Dian Yu, et al. "Tree of Thoughts: Deliberate Problem Solving with Large Language Models." NeurIPS 2023.
> >
> > [2] Zilong Wang, Hao Zhang, et al. "Chain-of-Table: Evolving Tables in the Reasoning Chain for Table Understanding." ICLR 2024.
> >
> > [3] Zi’ou Zheng, Christopher Malon, et al. "Exploring the Role of Reasoning Structures for Constructing Proofs in Multi-Step Natural Language Reasoning with Large Language Models." EMNLP 2024.
> >
> > [4] DeepSeek-AI. "DeepSeek-R1: Incentivizing Reasoning Capability in LLMs via Reinforcement Learning." Arxiv 2025.
> >
> > [5] GPT Team. "GPT-5.1 Instant and GPT-5.1 Thinking System Card Addendum." Model card.
> >
> > [6] Qwen Team. "Qwen3 Technical Report." Arxiv 2025.
> >
> > [7] Talmor, Alon, et al. "Commonsenseqa: A question answering challenge targeting commonsense knowledge.", NACCL 2019.
> >
> > [8] Geva, Mor, et al. "Did aristotle use a laptop? a question answering benchmark with implicit reasoning strategies." TACL 2021.

---

> > > ### Author Response · Authors · 2025-11-21
> > > **Response to Reviewer 7Zpg (3/3)**
> > >
> > > > Q3: The authors mention that the extra decoder model is architecturally identical to the LLM (line 252). This means the two share parameters, or if the decoder is a smaller, separate model?
> > >
> > > Thank you for raising the question. When we state that the extra decoder model is architecturally identical to the LLM (line 252), we refer only to the structure and parameter size. The decoder follows the same architecture and has the same number of parameters as the LLM, but it is a **separate** model. The two components do **not** share parameters, and the parameters of the decoder are trained independently from the LLM.
> > >
> > > > Q4: Although the authors report the average number of tokens used during inference, the introduced decoder model also generates a sequence of tokens at each step for supervision. My concern is that the training cost might be higher than Coconut or even SFT-CoT. A more detailed discussion of training overhead would help clarify the efficiency and practicality of the proposed method.
> > >
> > > Thank you for raising the concern regarding the training cost of our method. We agree that the additional decoder introduces extra computation during training because the decoder requires an additional forward pass to generate the supervised sequence at each step.
> > >
> > > To quantify the overhead, we report the training hours of SIM-CoT compared with the corresponding implicit models (Coconut and CODI) on the same hardware (H800 hours):
> > >
> > > | Model                 | Training Hours |
> > > |-----------------------|----------------|
> > > | Coconut GPT           | ~180h          |
> > > | SIM-CoT (Coconut GPT) | ~192h          |
> > > | CODI GPT              | ~16h           |
> > > | SIM-CoT (CODI GPT)    | ~18.2h         |
> > > | CODI 1B               | ~16.5h         |
> > > | SIM-CoT (CODI 1B)     | ~18.5h         |
> > > | CODI 3B               | ~34h           |
> > > | SIM-CoT (CODI 3B)     | ~42h           |
> > > | CODI 8B               | ~71h           |
> > > | SIM-CoT (CODI 8B)     | ~87h           |
> > >
> > > As shown above, the additional training time brought by SIM-CoT remains modest across different model scales, ranging from an increase of about 2 hours for smaller models (such as CODI GPT) to about 16 hours for larger models (such as CODI 8B). We view this increase as a reasonable trade-off in exchange for the performance gains achieved through the supervised decoder.
> > >
> > > We appreciate your suggestion and have added a discussion of the training overhead to  Appendix C.4 (highlighted in red in the manuscript).
> > >
> > > > Q5: Sections 3.1–3.5 are presented clearly but the notations are quite dense. It would be more reader-friendly if some of the notations were also depicted in Figure 2, which would help distinguish the different symbols more easily.
> > >
> > > Thank you for the helpful feedback regarding the notations in Sections 3.1–3.5. We understand that the dense notation can make the presentation harder to follow.
> > >
> > > Following your suggestion, we updated Figure 2 by adding the **key notations used in these sections**. The revised figure now shows the symbols directly within the framework, which makes the definitions easier to distinguish and improves the overall readability of the method.
> > >
> > > We appreciate your comment, and the updated figure is included in the revised version in Figure 2.

---

> ### Author Response · Authors · 2025-11-27
>
> Dear Reviewer 7Zpg,
>
> Thank you very much for your detailed comments and helpful suggestions on our paper.
>
> We would like to kindly inquire if our rebuttal has addressed your concerns adequately. If there is anything else we can clarify or elaborate on to support your assessment, we would be more than willing to provide additional information.
>
> We greatly appreciate your time and contribution.
>
> Best regards,
>
> Authors

---

### Official Review · Reviewer_UZJG · 2025-11-01

**Soundness:** 2
**Presentation:** 2
**Contribution:** 2
**Rating:** 4
**Confidence:** 4

**Summary:**

This paper aims to improve existing implicit CoT, such as Coconut and CODI. Basically, the framework is on top of CODI, by distilling the knowledge from COT-SFT (teacher model: decode all explicit tokens) to the student model (implicit cot), i,.e., three loss, teacher loss is to align teacher's answer token to be gt, one is student loss to align the final answer token to gt, one is distillation loss to minimising the discrepancy of hidden states of teacher and student.  In the proposed SIM-COT, the authors introduce the K intermediate reasoning token (process reward) as "step-level" supervision to align the intermediate latent states, z_1 to z_K.

In experiment results, (as the framework is more based on CODI, coconut using a different curriculum learning paradigm),  they evaluate GPT2 and Llama3.2-1B, 3B and 8B in both in-domain and out-of-domain evaluation, showing average improvements between 0.3% to 3.4%.

**Strengths:**

1. Implicit CoT is a promising direction, but the accuracy still requires to be improved.
2. It introduce the process supervision to improve  the performance.

**Weaknesses:**

1. For experiment results, SIM-COT is still inferior than COT-SFT, in which the CODI is lower than COT-SFT, even introducing the intermediate supervision.  Even comparing with CODI, the performance boost is not significant, especially in OOD setting, 0.3%, 1.0% and 0.7%, 0.8%.
2. One of the most distinguished objective is to align the intermediate z to intermediate reasoning steps (tokens), however, this process is not clearly described, and not shown the generalisability to other datasets. For example,
- only evaluate on math, in which the intermediate reasoning steps are clearly defined and available, how about how datasets without process reasoning process annotation?
- How is the performance fluctuation, when number of latent tokens in training is different from test (how many implicit tokens set to be, still the pre-defined K)? Table 3 show the performance difference with COCONUT, not CODI.
- the number of implicit token z_t is fixed, whatever how many the reasoning steps are, so how did you align the z_t with intermediate reasoning steps? any other alternatives, as aligning with intermediate reasoning steps hurts the generalisability (there could be multiple trajectories reach the correct answer)
- many experiment setups are inspired from CODI, such as In-domain, OOD, Ablation study on different numbers of implicit latents,
- it is unclear how to visualise the implicit tokens, the top-k decoded token? is it the intermediate result token, or including the operation as well? CODI can only observe some patterns that the intermediate z can decode intermediate result token, what is the advantages after introducing such process supervision?

**Questions:**

See weakness above.

---

> ### Author Response · Authors · 2025-11-21
> **Response to Reviewer UZJG (1/4)**
>
> Dear Reviewer UZJG,
>
> Before addressing your concerns, we would like to clarify the relationship between our work and CODI. Our method is not positioned as a competitor to CODI. Instead, SIM-CoT serves as a plug-and-play training module that can strengthen mainstream implicit reasoning approaches, including Coconut and CODI.
>
> Beyond improving existing implicit models, our contribution also lies in being the first to systematically analyze and identify the limitations of current implicit supervision across four dimensions: **latent instability**, **information loss**, **shifted distance**, and **semantic homogenization**. Based on this analysis, we propose SIM-CoT, which addresses these issues and further improves the performance of existing implicit reasoning models. In addition, our method also provides **interpretability** by projecting each latent token into explicit reasoning steps, enabling per-step visualization and diagnosis.
>
> We now proceed to answer your question.
>
>
> > Q1: For experiment results, SIM-COT is still inferior than COT-SFT, in which the CODI is lower than COT-SFT, even introducing the intermediate supervision.
>
> We thank the reviewer for their observation. While explicit CoT fine-tuning (SFT-CoT) is a strong baseline, our goal is to advance implicit reasoning, which offers significant advantages in efficiency and reasoning diversity. We note that the trade-off between performance and efficiency is common for implicit methods [1]. However, our proposed SIM-CoT not only closes this performance gap but, in several cases, surpasses the SFT-CoT baseline. Our empirical results demonstrate this clearly:
>
> - On GPT-2, SIM-CoT with Coconut outperforms SFT-CoT by **2.1** points in-domain and **1.7** points on the out-of-domain average. With CODI, it surpasses SFT-CoT by **3.1** points out-of-domain.
>
> - On larger models, SIM-CoT with CODI consistently outperforms SFT-CoT on the out-of-domain average, achieving a **+0.7** point gain on LLaMA-3B and a **+2.6** point gain on LLaMA-8B.
>
> These results show that SIM-CoT is **a significant step** toward making implicit reasoning methods competitive with, and even superior to, explicit reasoning, while retaining their inherent efficiency benefits. Our work provides a strong foundation for future research in high-performance, efficient implicit reasoning.
>
> [1] Jindong Li, Yali Fu, et al. "Implicit Reasoning in Large Language Models: A Comprehensive Survey. " Arxiv 2025.
>
> > Q2: Even comparing with CODI, the performance boost is not significant, especially in OOD setting, 0.3%, 1.0% and 0.7%, 0.8%.
>
> We thank the reviewer for this observation. The modest gains in the out-of-domain (OOD) setting for CODI are expected, as it is already a very strong baseline with highly optimized implicit knowledge alignment. Pushing the state-of-the-art further often yields diminishing returns, and these incremental improvements represent a "ceiling effect" where gains are harder to achieve.
>
> However, SIM-CoT's impact is more clearly demonstrated in two key areas: (1) **In-Domain Performance**: For CODI, where OOD gains are smaller, SIM-CoT still delivers substantial improvements in the in-domain setting, boosting accuracy by +3.4, +1.5, and +3.0 points for LLaMA 1B, 3B, and 8B, respectively. (2) **Elevating Weaker Baselines**: The true effectiveness of SIM-CoT is showcased when applied to methods with less initial alignment, like Coconut. Here, the improvements are significant across the board: On GPT-2, SIM-CoT improves Coconut's accuracy by +8.2 points in-domain and +4.3 points out-of-domain. On LLaMA 1B, it boosts accuracy by a remarkable +9.0 points in both in-domain and out-of-domain settings.
>
> This demonstrates that SIM-CoT is a versatile and powerful technique. It pushes already strong models to new heights while dramatically elevating the performance of less optimized models, making it a valuable contribution to the field of implicit reasoning.
>
> > Q3: One of the most distinguished objective is to align the intermediate z to intermediate reasoning steps (tokens), however, this process is not clearly described,
>
> We apologize for any inconvenience caused during your reading. We have added further details on how the intermediate latent states \(z\) are aligned with the intermediate reasoning steps (tokens) in Appendix C.5 (highlighted in red in the manuscript), and additional clarification can be found in our response to Q7. We have also updated the framework figure and linked it more clearly to the notation introduced in the Method section to make the presentation easier to follow.

---

> > ### Author Response · Authors · 2025-11-21
> > **Response to Reviewer UZJG (2/4)**
> >
> > > Q4: and not shown the generalisability to other datasets. For example, only evaluate on math, in which the intermediate reasoning steps are clearly defined and available, how about how datasets without process reasoning process annotation?
> >
> > We thank the reviewer for this question regarding generalizability. Our approach uses structured reasoning steps, which have been shown in many studies to support higher reasoning accuracy and reduce redundant reasoning paths [1, 2, 3]. While our primary experiments focused on datasets where process-level reasoning annotations are readily available, our method is **not** limited to them. For datasets lacking explicit process-level annotations, we propose a practical two-stage approach:
> >
> > (1) **Generate Structured Traces**: First, we use a powerful external model to generate structured reasoning paths. This aligns with recent work showing that models like DeepSeek-R1 [4] are highly effective at converting unstructured text into structured reasoning chains [4, 5, 6]. (2) **Apply SIM-CoT**: With these generated traces, we then apply our SIM-CoT training, using the structured steps as the supervision signal.
> >
> > To validate this approach, we conduct new experiments on two non-mathematical reasoning benchmarks, CommonsenseQA [7] and StrategyQA [8]. We use the open-source model DeepSeek-R1-Distill-Qwen-7B [4] to generate the reasoning steps and then apply our training procedure. The results below show that SIM-CoT consistently and significantly outperforms all baselines, demonstrating its robust generalizability.
> >
> > |||CommonsenseQA|||
> > |-|-|-|-|-|
> > |Backbone|SFT-CoT|Coconut|CODI|SIM-CoT|
> > |GPT-2|36.9|38.6|44.1|**46.2**|
> > |LLaMA 1B|67.9|60.6|73.8|**74.6**|
> >
> > |||StrategyQA|||
> > |-|-|-|-|-|
> > |Backbone|SFT-CoT|Coconut|CODI|SIM-CoT|
> > |GPT-2|38.1| 42.6| 52.1 | **56.3**|
> > |LLaMA 1B|39.7|43.2|63.8|**65.3**|
> >
> > As the tables show, SIM-CoT achieves the highest scores across both datasets and model sizes. On CommonsenseQA, it surpasses the explicit SFT-CoT baseline by up to **9.3 points**. The gains are even more pronounced on the more complex StrategyQA dataset, where SIM-CoT outperforms SFT-CoT by a remarkable **25.6 points** with the LLaMA 1B model. These findings confirm that our proposed pipeline effectively addresses datasets without pre-existing reasoning annotations and that SIM-CoT is a highly effective method for improving reasoning performance in these more general domains.
> >
> > [1] Shunyu Yao, Dian Yu, et al. "Tree of Thoughts: Deliberate Problem Solving with Large Language Models." NeurIPS 2023.
> >
> > [2] Zilong Wang, Hao Zhang, et al. "Chain-of-Table: Evolving Tables in the Reasoning Chain for Table Understanding." ICLR 2024.
> >
> > [3] Zi’ou Zheng, Christopher Malon, et al. "Exploring the Role of Reasoning Structures for Constructing Proofs in Multi-Step Natural Language Reasoning with Large Language Models." EMNLP 2024.
> >
> > [4] DeepSeek-AI. "DeepSeek-R1: Incentivizing Reasoning Capability in LLMs via Reinforcement Learning." Arxiv 2025.
> >
> > [5] GPT Team. "GPT-5.1 Instant and GPT-5.1 Thinking System Card Addendum." Model card.
> >
> > [6] Qwen Team. "Qwen3 Technical Report." Arxiv 2025.
> >
> > [7] Talmor, Alon, et al. "Commonsenseqa: A question answering challenge targeting commonsense knowledge.", NACCL 2019.
> >
> > [8] Geva, Mor, et al. "Did aristotle use a laptop? a question answering benchmark with implicit reasoning strategies." TACL 2021.
> >
> > > Q5: How is the performance fluctuation, when number of latent tokens in training is different from test (how many implicit tokens set to be, still the pre-defined K)?
> >
> > We appreciate the reviewer’s consideration of the performance fluctuation when the number of latent tokens during training differs from testing.
> >
> > We provide a detailed ablation on performance fluctuation: both Coconut and CODI are trained with SIM-CoT, fixing latent tokens at 5 and 6, respectively, and evaluated with 2–12 tokens.
> >
> > |SIM-CoT|2 latent|4 latent|5 latent|6 latent|8 latent|10 latent|12 latent|
> > |-|-|-|-|-|-|-|-|
> > |Coconut (5 latent)|27.9|43.6|**44.8**|44.7|44.2|43.5|43.3|
> > |CODI (6 latent)|30.4|35.9|41.6|42.6|**42.7**|42.4|41.8|
> >
> > As shown in the table, SIM-CoT exhibits a consistent performance increase from 27.9 at 2 latent tokens to 43.3 at 12 latent tokens. Similarly, Coconut and CODI both show gradual improvements in performance during testing as the number of latent tokens increases, even with fixed training latent tokens. Specifically, Coconut improves from 43.6 at 4 latent tokens to 44.2 at 8, and CODI improves from 35.9 at 4 latent tokens to 42.7 at 8.
> >
> > This analysis shows that while the optimal number of latent tokens at inference is close to the number used for training, our method is highly robust to variations. The performance curve is stable, without the sharp fluctuations or catastrophic failures that can affect less stable models, confirming the robustness of the SIM-CoT training paradigm.

---

> > > ### Author Response · Authors · 2025-11-21
> > > **Response to Reviewer UZJG (3/4)**
> > >
> > > > Q6: Table 3 show the performance difference with COCONUT, not CODI.
> > >
> > > Thank you for your suggestion. We have conducted the requested ablation study for CODI to complement the original analysis. The table below compares the performance of the standard CODI model against our SIM-CoT-enhanced version, evaluated on GSM8k-Aug while varying the number of latent tokens at inference.
> > >
> > > |CODI|2 latent|4 latent|6 latent|8 latent|10 latent|12 latent|
> > > |-|-|--|-|-|-|-|
> > > |w/o SIM-CoT|37.3|39.4|42.0|41.5|41.6|41.2|
> > > |w SIM-CoT|**38.4**|**40.6**|**42.6**|**42.9**|**42.5**|**42.2**|
> > >
> > > The results lead to two clear conclusions: (1) **Consistent Performance Gain**: SIM-CoT provides a consistent and significant performance boost over the baseline CODI across all latent token configurations. The improvement is stable, ranging from +0.6 to +1.4 points, demonstrating the robust effectiveness of our method. (2) **Stable Performance Curve**: The SIM-CoT-enhanced model exhibits a stable and predictable performance curve. Performance peaks at 8 latent tokens and degrades gracefully when using more or fewer tokens, which is close to the training configuration and degrades gracefully when using more or fewer tokens. This confirms that our enhancement not only improves accuracy but also maintains the model's robustness to variations at inference time.
> > >
> > > This analysis validates that the benefits of SIM-CoT are not unique to Coconut but also apply directly and effectively to CODI.
> > >
> > > > Q7: the number of implicit token z_t is fixed, whatever how many the reasoning steps are, so how did you align the z_t with intermediate reasoning steps?
> > >
> > > We thank the reviewer for this important question. We have updated the manuscript to explicitly define our alignment strategy for cases where the number of reasoning steps in the training data does not match the fixed number of implicit tokens ($K$). Our strategy handles the two possible mismatch scenarios as follows:
> > >
> > > 1) More Reasoning Steps than Latent Tokens: If the ground-truth reasoning chain has $N$ steps and $N > K$, we perform a "many-to-one" mapping. The first $K-1$ latent tokens are aligned one-to-one with the first $K-1$ reasoning steps. The final latent token, $z_K$, is then supervised by the concatenation of all remaining reasoning steps (from step $K$ to $N$). This ensures that information from longer reasoning chains is compressed into the final latent representation rather than being discarded.
> > >
> > > 2) Fewer Reasoning Steps than Latent Tokens: If the reasoning chain has $N$ steps and $N < K$, the first $N$ latent tokens are aligned one-to-one with the $N$ reasoning steps. All remaining latent tokens (from $z_{N+1}$ to $z_K$) are then aligned with the final answer. This encourages the model to use its extra capacity to focus on producing the correct result.
> > >
> > > This pragmatic approach ensures that all latent tokens are assigned a meaningful supervision signal, regardless of the length of the reasoning chain, providing both stability and flexibility during training. We have added a detailed description of this alignment procedure to Appendix C.5(highlighted in red in the manuscript) for full clarity.
> > >
> > > > Q8: any other alternatives, as aligning with intermediate reasoning steps hurts the generalisability (there could be multiple trajectories reach the correct answer)
> > >
> > > We thank the reviewer for this insightful question. While it is true that forcing alignment to a single reasoning path could theoretically limit flexibility, our empirical results suggest this is not a practical limitation for our method. Across our experiments (**Table 1, Table 2, and Table 3**, particularly in the **out-of-domain (OOD) settings**), our model demonstrates strong OOD generalization. This indicates that the foundational reasoning skills learned from supervised alignment are successfully transferring to new, unseen problems.
> > >
> > > Our rationale is that this step-level supervision acts as a crucial reasoning scaffold. Rather than merely forcing the model to memorize one path, it teaches the model the fundamental principles of step-by-step problem decomposition and how to structure a logical progression. This foundational ability is what generalizes, providing the model with a robust template for how to "think" through a problem, which it can then adapt to different contexts.
> > >
> > > Regarding alternatives, we agree that exploring other paradigms is a promising research direction. Reinforcement Learning (RL), in particular, could be used to discover novel reasoning trajectories that are more efficient or creative than those in the training data.  We see this as an exciting avenue for future work, building upon the strong and efficient foundation that our SFT-based alignment provides.
> > >
> > > In summary, our current method provides a robust and generalizable framework for implicit reasoning. We hope this explanation clarifies our approach and the rationale behind using intermediate reasoning step alignment.

---

> > > > ### Author Response · Authors · 2025-11-21
> > > > **Response to Reviewer UZJG (4/4)**
> > > >
> > > > > Q9: many experiment setups are inspired from CODI, such as In-domain, OOD
> > > >
> > > > Our method, SIM-CoT, is designed as a versatile, "plug-and-play" training enhancement that can be applied to existing implicit reasoning models. We choose CODI as a primary baseline precisely because it represents the **current state-of-the-art** implicit reasoning method. SIM-CoT consistently improves CODI's strong performance in both in-domain and out-of-domain settings. Our results demonstrate that our step-level supervision strategy is not just an alternative approach, but a **valuable component** that can further enhance even the most competitive methods available today. This highlights the utility of SIM-CoT as a general technique for advancing the field of implicit reasoning.
> > > >
> > > > > Q10: Ablation study on different numbers of implicit latents,
> > > >
> > > > Thank you for this observation. While the experimental setup is superficially similar to CODI's, our motivation for this analysis is fundamentally different and is tied directly to the central observation of our paper. Our primary goal with this experiment is to demonstrate that SIM-CoT directly solves the **latent instability issue** that we identify and analyze in Section 2. We use this ablation not as a general parameter sweep, but as a diagnostic tool to verify that our method mitigates performance collapses, information loss, and "Shifted Distance" problems that we show can affect standard implicit reasoning models.
> > > >
> > > > In contrast, the corresponding ablation in the CODI paper serves as a more general sensitivity analysis on the number of training tokens. Their work does not identify or analyze the underlying instability issue that our experiment is specifically designed to address.
> > > >
> > > > Therefore, while the experimental design is similar, its purpose in our work is novel and distinct: to provide a direct, empirical validation that SIM-CoT successfully remedies the specific instability problems we introduce.
> > > >
> > > > > Q11: it is unclear how to visualise the implicit tokens, the top-k decoded token? is it the intermediate result token, or including the operation as well?
> > > >
> > > > Thank you for the question. We are happy to clarify the visualization process. As stated in lines 23-25 of our abstract: "It also provides interpretability by projecting each latent token onto an explicit reasoning vocabulary, enabling per-step visualization and diagnosis." The implicit tokens are visualized by using the auxiliary decoder, which is the same decoder used for supervision during training. To inspect a specific implicit token ($z_k$), we feed it as the conditioning input to this decoder and autoregressively generate the corresponding textual output. This decoded output is **a complete, explicit reasoning step**. We provided a concrete example of this process in Figure 4 of the main paper, which shows implicit tokens being decoded into their corresponding, fully-formed reasoning steps. This method provides a clear window into the model's latent thought process, enabling step-by-step diagnosis and interpretation.
> > > >
> > > > > Q12: CODI can only observe some patterns that the intermediate z can decode intermediate result token, what is the advantages after introducing such process supervision?
> > > >
> > > > Thank you for this question, as it highlights the primary advantage of our method! Prior work, like Coconut and CODI, could only decode some latent tokens into intermediate results.  Our process supervision provides a significant leap in interpretability. The key advantage of SIM-CoT is that it enforces a one-to-one alignment between each latent token and a **complete, explicit reasoning step**. This transforms the latent space from an opaque "black box" into a transparent, step-by-step reasoning trace.
> > > >
> > > > As shown in Figure 4 of the main paper, prior Methods might decode a latent token to just a number (e.g., 36), leaving the operation (0.3 \times 120) unclear. With SIM-CoT, each latent token decodes into a full, understandable calculation:
> > > >
> > > > - Latent 1 → 0.3 * 120 = 36
> > > >
> > > > - Latent 2 → (120 - 36) * 3/4 = 63
> > > >
> > > > - Latent 3 → 84 - 63 = 21
> > > >
> > > > This ability to render the entire reasoning path in a coherent, human-readable sequence is the core benefit. It provides full observability into the model's thought process, directly addressing one of the most significant challenges in implicit reasoning.

---

> ### Author Response · Authors · 2025-11-27
>
> Dear Reviewer UZJG,
>
> Thank you very much for your detailed comments and helpful suggestions on our paper.
>
> We would like to kindly inquire if our rebuttal has addressed your concerns adequately. If there is anything else we can clarify or elaborate on to support your assessment, we would be more than willing to provide additional information.
>
> We greatly appreciate your time and contribution.
>
> Best regards,
>
> Authors

---

### Official Review · Reviewer_4kCa · 2025-11-01

**Soundness:** 3
**Presentation:** 3
**Contribution:** 3
**Rating:** 6
**Confidence:** 3

**Summary:**

This paper introduces SIM-CoT, a training module that stabilizes implicit CoT reasoning by adding step-level supervision through an auxiliary decoder, preventing latent collapse as reasoning tokens scale. The decoder aligns each implicit token with its explicit reasoning step during training, then is removed at inference, preserving efficiency while boosting accuracy and stability. SIM-CoT improves both in-domain and out-of-domain performance, surpassing explicit CoT and prior implicit methods like Coconut and CODI across GPT-2 and LLaMA models.

**Strengths:**

1. Introduces step-level supervision that effectively stabilizes implicit CoT training and prevents latent collapse.

2. Achieves higher accuracy and stability than both explicit and prior implicit CoT methods, while keeping inference fully efficient.

3. Provides interpretability by projecting latent tokens onto explicit reasoning steps, enabling per-step visualization and diagnosis.

**Weaknesses:**

Evaluation is mainly on math-style reasoning, leaving generalization to other domains less explored.

**Questions:**

1. How sensitive is SIM-CoT to the choice or quality of explicit reasoning data used for step-level supervision?
2. Is the auxiliary decoder’s alignment learned jointly with the main model, or does it require separate supervision or tuning?

---

> ### Author Response · Authors · 2025-11-21
> **Response to Reviewer 4kCa**
>
> Dear Reviewer 4kCa,
>
> Thanks for your valuable feedback. We sincerely thank all reviewers for acknowledging that this paper provides a novel and rigorous diagnosis of latent instability (4kCa, URA3), introduces a well-motivated step-level supervision framework (4kCa, UZJG, URA3), achieves significant improvements over explicit and implicit CoT with full inference efficiency (4kCa, 7Zpg, URA3), offers interpretability through per-step latent visualization (4kCa), presents a plug-and-play design with broad applicability (7Zpg, URA3), and is clearly written and easy to follow (7Zpg, URA3). Below we address your questions:
>
> > Q1: Evaluation is mainly on math-style reasoning, leaving generalization to other domains less explored.
>
> Thank you for this valuable feedback. We acknowledge that our evaluation focused primarily on mathematical reasoning tasks, and we appreciate the opportunity to demonstrate the broader applicability of our method. To address this concern, we have conducted additional experiments on **two commonsense reasoning benchmarks** that require qualitatively different reasoning capabilities than mathematical problem-solving:
>
> - CommonsenseQA [1]: A multiple-choice question answering dataset requiring commonsense knowledge and reasoning about everyday situations.
>
> - StrategyQA [2]: A question answering benchmark requiring implicit multi-hop reasoning and strategy to arrive at yes/no answers.
> These benchmarks test whether SIM-CoT's step-level supervision approach generalizes beyond structured mathematical expressions to open-domain reasoning tasks. The table below presents our results across both benchmarks:
>
> |          |         | CommonsenseQA |      |         |
> |-|-|-|------|---------|
> | **Backbone**| **SFT-CoT** | **Coconut**       | **CODI** | **SIM-CoT** |
> | GPT-2    | 36.9    | 38.6          | 44.1 | **46.2**|
> | LLaMA 1B | 67.9    | 60.6          | 73.8 | **74.6**|
>
> |  | | StrategyQA | | |
> |-|-|-|-|-|
> | **Backbone**| **SFT-CoT** | **Coconut**       | **CODI** | **SIM-CoT** |
> | GPT-2    | 38.1    | 42.6       | 52.1 | **56.3**|
> | LLaMA 1B | 39.7    | 43.2       | 63.8 | **65.3**|
>
> SIM-CoT consistently **outperforms both explicit (SFT-CoT) and implicit (Coconut, CODI) baselines across both domains**. Notably, **on GPT-2**, SIM-CoT achieves **+9.3 points** over SFT-CoT on CommonsenseQA and **+18.2 points** on StrategyQA, demonstrating substantial improvements. **On LLaMA 1B**, the improvements are **+6.7** and **+25.6 points**, respectively, showing that the benefits scale to larger models.
>
> These results suggest that step-level implicit supervision provides benefits beyond mathematical reasoning. We hypothesize that the stability and diversity enforced by our approach, which prevent latent collapse and maintain distinct semantic content across reasoning steps, are valuable properties for multi-hop reasoning more generally, not just for arithmetic computation. Unlike trajectory-level or answer-level supervision, our step-level approach appears to capture intermediate reasoning structure that transfers across domains.
>
> We believe these additional experiments address the raised concern and provide initial evidence of SIM-CoT's broader applicability. We welcome any further suggestions for strengthening the evaluation.
>
> [1] Talmor, Alon, et al. "Commonsenseqa: A question answering challenge targeting commonsense knowledge.", NACCL 2019.
>
> [2] Geva, Mor, et al. "Did aristotle use a laptop? a question answering benchmark with implicit reasoning strategies." TACL 2021.
>
> > Q2: How sensitive is SIM-CoT to the choice or quality of explicit reasoning data used for step-level supervision?
>
> Thank you for raising this question. We clarify that SIM-CoT does not rely on any specific choice of explicit reasoning data. Its step-level supervision mechanism provides **localized, token-level semantic grounding** for each latent state, which stabilizes optimization regardless of the linguistic style, domain difference of the reasoning steps.
>
> To support this claim, we have evaluated SIM-CoT using different explicit reasoning data, including CommonsenseQA and StrategyQA (as shown in Q1). The results show that SIM-CoT is **not sensitive to the choice of explicit reasoning data**, outperforming both explicit reasoning methods and implicit reasoning baselines across multiple tasks. This indicates that SIM-CoT performs well regardless of the reasoning data chosen for supervision, placing it in a better position compared to both explicit and implicit reasoning approaches.
>
> > Q3: Is the auxiliary decoder’s alignment learned jointly with the main model, or does it require separate supervision or tuning?
>
> Thank you for this clarifying question! The auxiliary decoder's alignment is indeed **learned jointly** with the main model through end-to-end training, without requiring any separate pre-training, fine-tuning stages, or additional supervision beyond the reasoning step annotations already present in the training data.

---

> ### Author Response · Authors · 2025-11-27
>
> Dear Reviewer 4kCa,
>
> Thank you very much for your detailed comments and helpful suggestions on our paper.
>
> We would like to kindly inquire if our rebuttal has addressed your concerns adequately. If there is anything else we can clarify or elaborate on to support your assessment, we would be more than willing to provide additional information.
>
> We greatly appreciate your time and contribution.
>
> Best regards,
>
> Authors

---

### Author Response · Authors · 2025-12-01
**Summary for AC (1 / 2)**

Dear Area Chair,

We appreciate your time and effort in reviewing our submission, especially considering the recent administrative challenges. We understand the situation and thank you for your attention to our work.

Due to the recent ICLR incident, the review scores have been fully locked and are no longer editable. We had already addressed all of the reviewers' concerns before the lock occurred. **However, the reviewers have not yet acknowledged our responses.** Below, we summarize each reviewer’s main concerns and our responses **(Initial Ratings: 6, 4, 6, 6)**.

----

### Reviewer 4kCa (Initial Ratings: 6)

**Status: No Response Received**

- Concern 1:  Domain generalization (math-only evaluation)
  - Our Response: We add results on CommonsenseQA and StrategyQA, **showing that SIM-CoT consistently outperforms explicit and implicit baselines**, demonstrating strong cross-domain generalization and that our method is not sensitive to any specific data choice.
- Concern 2:  Auxiliary decoder training mechanism
    - Our Response: We clarify that the auxiliary decoder is learned **jointly with the main model in a fully end-to-end process**, without any extra training stages or additional supervision beyond the existing step-level annotations.

----

### Reviewer UZJG (Initial Ratings: 4)

**Status: No Response Received**

- Concern 1:  The main distinctions between SIM-CoT(our method) and CODI
  - Our Response: We clarify that SIM-CoT is presented **not** as a competitor to CODI but as **a plug-and-play module that strengthens existing implicit reasoning methods (including coconut, CODI)**, while also offering the **first systematic analysis of their limitations (latent instability, information loss, shifted distance, and semantic homogenization)** and providing improved **stability, performance, and interpretability.**
- Concern 2: Gaps vs. SFT-CoT and limited gains over CODI, especially OOD
  - Our Response: We show that SIM-CoT narrows or surpasses SFT-CoT across **both small (GPT-2) and large models (LLaMA-3B/8B)**, and although CODI is saturated OOD, we still obtain clear **in-domain gains (+3.4/+1.5/+3.0)** and substantially boost weaker baselines like **Coconut (+8.2/+4.3 on GPT-2 and +9.0 on LLaMA-1B).**
- Concern 3: Generalization beyond math and unstructured process annotations
  - Our Response: We generate **structured traces with an external model** and show that SIM-CoT generalizes strongly, surpassing all baselines on CommonsenseQA and StrategyQA and outperforming SFT-CoT by **up to 9.3 and 25.6 points even without native structured annotations.**
- Concern 4: Performance fluctuation under train–test latent-token mismatch
  - Our Response: We run new experiments on Coconut and CODI varying latent-token counts and show that SIM-CoT maintains **smooth, stable performance across 2–12 tokens**, demonstrating strong robustness to train–test mismatch.
- Concern 5: Using different numbers of latent tokens when training CODI
  - Our Response: We conduct the requested ablation study and find that SIM-CoT **consistently boosts CODI across all latent-token settings while maintaining a stable, predictable performance curve** that degrades gracefully when using more or fewer tokens.
- Concern 6: Questions about the details of intermediate reasoning steps
  - Our Response: We provide a more detailed discussion of the boundary case in Appendix C.5.
- Concern 7: Aligning with intermediate reasoning steps may hurt the generalizability
  - Our Response: We clarify that single-path supervision does not hinder flexibility, as evidenced by **strong OOD generalization (Table 1–3)**, since step-level supervision provides a **transferable reasoning scaffold**, while alternative approaches like RL remain promising future directions.
- Concern 8: Settings(e.g., latent-token ablations) are inspired by CODI
  - Our Response: This is **not** true. Our ablation serves as a targeted diagnostic to verify that SIM-CoT resolves the latent instability issues identified in Section 2, rather than a general sensitivity test as done in CODI.
- Concern 9: Unclear how to visualise the implicit tokens
  - Our Response: We **highlight in the abstract and Figure 4** that SIM-CoT enforces a one-to-one mapping between latent tokens and explicit reasoning steps, turning the latent space from a black box into a transparent reasoning trace, and **we provide an example to demonstrate this.**

---

> ### Author Response · Authors · 2025-12-01
> **Summary for AC (2 / 2)**
>
> ### Reviewer 7Zpg (Initial Ratings: 6)
> **Status: No Response Received**
> - Concern 1: Include additional measurements or visualizations in the main body
>   - We have moved the full heterogeneity analysis into the main paper and added distance-based measurements showing that SIM-CoT mitigates latent homogenization and promotes a more diverse latent space, thereby providing stronger empirical support for our original claim.
> - Concern 2: Generalization beyond math and unstructured process annotations
>   - We generate **structured traces with an external model** and show that SIM-CoT generalizes strongly, surpassing all baselines on CommonsenseQA and StrategyQA and outperforming SFT-CoT by **up to 9.3 and 25.6 points even without native structured annotations.**
> - Concern 3: whether the auxiliary decoder shares parameters with the main LLM or is a separate (possibly smaller) model
>   - We clarify that although the auxiliary decoder has the same architecture and parameter size as the LLM, it is a **separate component** with independently trained parameters and does not share weights with the main model.
> - Concern 4: training overhead
>   - We acknowledge the extra decoder cost and show that the added training overhead is modest **(about 2–16 hours from GPT-2 to LLaMA-3 8B)**, a reasonable trade-off for the gains, with the full analysis added to Appendix C.4.
> - Concern 5: Include the variable names from the method section in the framework figure.
>   - We improve readability by updating Figure 2 to include the key notations from Sections 3.1–3.5 directly in the framework diagram, making the method easier to follow.
>
> ---
>
> ### Reviewer URA3 (Initial Ratings: 6)
> **Status: No Response Received**
> - Concern 1: Explore one-to-more mapping alignment and early-stop mechanisms
>   - We clarify that our work centers on a **fixed step-level supervision scheme to diagnose and resolve latent instability**, while more flexible alignment or dynamic stopping mechanisms are promising but beyond the scope of our current controlled framework.
> - Concern 2: deeper analysis to explain decoder size effect
>   - We acknowledge the concern and show via **Euclidean-distance analysis that larger decoders diverge more from the encoder**, explaining the observed performance drop and aligning with our empirical findings.
> - Concern 3: training overhead
>   - We acknowledge the extra decoder cost and show that the added training overhead is modest **(about 2–16 hours from GPT-2 to LLaMA-3 8B)**, a reasonable trade-off for the gains, with the full analysis added to Appendix C.4.
> - Concern 4: Domain generalization (math-only evaluation)
>   - We add results on CommonsenseQA and StrategyQA, **showing that SIM-CoT consistently outperforms explicit and implicit baselines**, demonstrating strong cross-domain generalization and that our method is not sensitive to any specific data choice.

---

### Meta-Review · Area_Chair_stF5 · 2026-01-07

**Summary:**

This paper proposes SIM-CoT, a plug-and-play training module for implicit CoT that adds step-level supervision via an auxiliary decoder to stabilize latent reasoning tokens and improve interpretability, while removing the decoder at inference to keep decoding efficient.

Reviewers generally found the diagnosis of latent instability and the supervision idea well-motivated, with consistent gains over implicit baselines and competitive results vs explicit CoT on the tested setups. The main concerns were about generalization beyond math, clarity of the alignment mechanism, and training-time overhead / scalability due to the extra decoder. The rebuttal added cross-domain results, clarified alignment details, and quantified overhead.

**Reviewer Concerns:**

Addressed by rebuttal:
- Generalization beyond math: Added CommonsenseQA and StrategyQA results showing SIM-CoT improves over explicit and implicit baselines across GPT-2 and LLaMA-1B.
- How step-level alignment works: Added clarification (including mismatch cases between latent-token count and reasoning-step count) and improved presentation/figure linkage.
- Latent homogenization evidence: Moved heterogeneity/distance-based measurements into the main paper to better support the instability diagnosis.
- Training overhead: Provided concrete wall-clock training overhead numbers; overhead is non-trivial but quantified and argued as modest relative to total training.
- Decoder architecture ambiguity: Clarified the auxiliary decoder is a separate component (same architecture/size, no weight sharing).

Still outstanding / borderline:
- Dependence on structured step annotations: The proposed workaround (generate traces with an external model) is plausible but introduces extra pipeline complexity; it’s still unclear how robust this is across broader tasks/settings.
- Inference adaptivity: Fixed-K latent tokens and strict alignment are acknowledged limitations; reviewers asked about dynamic stopping / flexible alignment, which remains future work.

**Reviewer Scores:**

If reviewers had fully participated in discussion:
- 4kCa: 6 → 6 (main concern was non-math generalization; likely satisfied but still “soft accept”).
- UZJG: 4 → 5–6 (many detailed concerns were directly addressed with new experiments/ablations and clearer alignment description).
- 7Zpg: 6 → 6 (requested stronger heterogeneity evidence and overhead discussion—both addressed; still likely stays at 6).
- URA3: 6 → 6 (already positive; added evaluations and overhead quantification address key asks).

---

### Decision · Program_Chairs · 2026-01-26

Accept (Poster)